# Institutional Determinants of Budgetary Expenditures. A BMA-Based Re-Evaluation of Contemporary Theories for OECD Countries

**Krzysztof Beck** [1,†] and **Michał Możdżeń** [2,*,†]

1    Department of Econometrics, Lazarski University in Warsaw, 02-662 Warszawa, Poland;
     beckkrzysztof@gmail.com
2    Department of Public Policy, Cracow University of Economics, 31-510 Kraków, Poland
*    Correspondence: mozdzenm@uek.krakow.pl
†    These authors contributed equally to this work.

**Abstract:** The article tackles the problem of the most important institutional determinants of public expenditures. Within the traditions of public choice and institutional economics, it tests several theories ranging from the fiscal commons framework, Political Business/Budget Cycle (PBC) and path dependence to veto players theory. Its novelty compared to previous research stems from an attempt to test several theories simultaneously, dealing with model uncertainty by using sensitivity analysis within the Bayesian Model Averaging framework with a vast prior structure in terms of model, g and multicollinearity dilution priors. The results confirm several hypotheses tested in the area of fiscal management across the recent decades within the group of developed economies, giving especially strong support to the tragedy of the fiscal commons and path dependence concepts, while only partial support to veto players theory. In contrast, explanations based on political budget cycle (PBC) theory are dismissed. Among other interesting findings reported in the study, Scandinavian countries turn out to be the most fiscally responsible when other institutional factors are taken into account. Similarly, contrary to other recent research into the issue of EU fiscal institutional framework, Euro area countries are characterized by limited public expenditures.

**Keywords:** Bayesian model averaging; fiscal commons; fiscal rules; Government expenditures; political institutions

## 1. Introduction

The problem of explaining the differences in public expenditures and their dynamics among democratic states is an old one, dating at least a century with posing of so-called Wagner's law [1]. From the practical standpoint, the problem is important, as public expenditures are linked with the magnitude of public deficits and debts, and indirectly affect fiscal sustainability. This issue in turn is still considered one of the most important problems with regards to risks of financial crises, although it lost some prominence in recent years in favor of external imbalances of a country generally, regardless of the legal constraints put on many countries in terms of their fiscal flexibility or some cases in which the crisis is connected directly to the level of public debt-such as in Greece [2]. It is being actively pursued across disciplines of political science and economics [3]. This issue has seen a variety of explanations from both disciplines ranging from empirical [4] to theoretical ones [5,6]. The most prominent explanation for the differences in public expenditures is connected to the concept of the "tragedy of the fiscal commons" attributed to the problem first posed by Hardin [7] who tried to explain

the overexploitation of common natural resources. This issue was most vividly described in the area of public finance by [8]. The basic premise of the concept relates to the positive relationship between expenditures and the number of political actors that are able to "tap into" the fiscal resource [9,10]. Over recent decades, this problem has been tackled by several scholars working within the framework of public choice theory and new institutional economics [11]. In this vein of research, institutional features of the political system which put constraints on the discretionary powers of particular actors contribute to controlling the size and dynamics of public expenditures as well as working against the so-called "deficit bias" of the democratic process. This is the tradition which this article is also strongly linked to. There are, of course, different explanations for the dynamics of the budgetary process (and public finance in general), which include e.g., political ideology (Veto Players Theory-[12]), internal power struggles within the political system (Selectorate Theory-[13]) or ease of access to the political sphere by external interest groups (Access Point Theory-[14], State Capture Theory-see [15]). One has to acknowledge an important vein of research into political budget cycles which deals with the dynamics of public expenditures and deficits with relation to electoral cycle [16–21]. Overall, however, it seems that the "tragedy of the fiscal commons" (ToFC) approach is the one with strongest theoretical and empirical basis to date.

The goal of the paper is to assess the explanatory power of most prominent theories connected to the problem of political economy of fiscal sustainability, using relatively novel (and still underused in the area of political economy) methodological approach called Bayesian Model Averaging and a dataset of important characteristics of OECD countries (and Bulgaria) compiled by the authors. The explanatory frameworks subjected to testing include: the tragedy of the fiscal commons [5], fiscal path dependency (e.g., [4]), veto players theory [12] and political budget cycles [17]. The novelty of the present article is based on the fact that it is working with a variety of hypotheses stemming from different enumerated theoretical approaches and strives to reevaluate them with the use of sensitivity analysis within the Bayesian model averaging framework.

The remainder of the paper is structured as follows. Section 2 presents literature review and comprises 3 subsections devoted to institutions of power legitimacy, institutions of internal power relations and budgetary rules. The data set used for estimation and Bayesian model averaging estimation strategy is described in Section 3. The estimation results are discussed in Section 4, while conclusions and discussion are provided in Section 5.

## 2. Literature Review and Research Hypotheses

The results of the research presented below are a part to the fast-growing empirical trend of research into the institutional determinants of the political process, in particular the budget process. However, recalling them will allow us to theoretically and empirically consolidate the current knowledge of the impact of institutions on the budget process. The state of current research will be presented in three categories: institutions of political power legitimacy, institutions of internal system relations and rules of the budget process.

### 2.1. Institutions of Power Legitimacy

Institutions of power legitimacy in established democracies should be understood as simply electoral institutions. Research into the applied rules of the emergence of power and their impact on the actions of political actors is relatively well developed and offers interesting conclusions.

Focusing on parliamentary elections, it should be noted that their institutional characteristics fundamentally differ in several dimensions. Political science analyzes in detail many of these dimensions, but they focus on three ones in particular: the range (size) of the district, the electoral formula, and the structure of electoral lists. This is important because, as [3] point out, these institutional dimensions are relatively frequently examined in relation to the size and structure of public spending (including budget expenditure).

It is generally assumed that in a system with large constituencies(districts), politicians put more emphasis on broad political programs, whereas in smaller ones they look for more specifically addressed proposals [22,23]. In connection with the need to gain the support of a larger part of society in large districts, some researchers suggest the existence of relatively higher total public expenditure in such systems [23], while others assume that the effect should be ambiguous [24]. The electoral formula decides how votes are converted into seats. In the majoritarian formula, only the candidates with the largest percentage of votes in the district receive seats. As part of the proportional formula, the number of seats depends on the percentage of votes obtained. Duverger's law states that in elections taking place in accordance with the majority formula, one should observe the emergence of stable two-party systems, and in the case of a proportional formula the political landscape should be more unstable and diverse, with high party and government fractionalization. Moreover, as [3,25] argue, in majoritarian systems we should expect a greater increase in the occurrence of political budget cycles than in the proportional or mixed formula due to the increased responsibility of politicians to their electorate. However, the PBC as a political phenomenon, is still subject to dispute, at least in the advanced countries, with some works showing its existence [26,27], while others increasingly subject it to rejection [17,28,29]. It must be noted that this phenomenon still seems prevalent at the sub-national level [30,31] or affects composition of spending [32], which we cannot consider in the paper.

The structure of electoral lists determines how citizens give their votes; by voting for individual candidates or for entire lists. As a rule, Reference [24], similarly to [33], predict that the list-voting system should generate a larger percentage of parliamentarians interested in abusing the system. On the other hand, increased loyalty to the party establishment at the expense of voters in a given district should reduce the interest of candidates who start in systems based on electoral lists (especially closed ones) in favoring the particular interest of their district [34].

Reference [35], as a result of research conducted among Latin American countries, stated that the problem of common fiscal resources is more visible in countries where electoral institutions emphasize personal accountability of parliamentarians. With this in mind, one can hypothesize that among democratic countries those with higher voter turnout should generate higher public expenditures [36].

## 2.2. Institutions of Internal Power Relations

In the case of the second institutional dimension, the degree of centralization/fragmentation of the budget process is a particularly often examined specificity. The dimension is most frequently operationalized in two ways, either with the help of the evaluation of relative strength of finance minister against line ministers, or by assessing the number of actors participating in the budget process. In general, the majority of research focuses on the impact of centralization on the size of budget surpluses/deficits; however, it seems that some conditional conclusions can also be drawn from them for the size of public expenditure. The negative impact of centralization on the size of budget deficits is indicated, among others, References [37–39] and on the basis of the analysis of the situation of individual countries [40]-Belgium, [41]-Sweden and [42]-Germany. A large scale historical analysis confirming the importance of centralization to fiscal prudence was done by [43]. In the context of Central and Eastern European countries, References [44,45] come to similar conclusions. Analogous results for other parts of the world were received by [46]-Latin American countries, [47]-Argentine provinces, and [42]-US states.

At the same time, in the context of the Veto Players theory, which combines institutional variables with ideological variables (both a larger number of VPs, greater ideological distance between VPs and greater stability of the ruling VP system, should have a negative impact on changes in the budget structure), research of [12,48] indicates that the structure of budgets is determined in dynamic terms by variables pointed to by theory—i.e., the more varied the ideological landscape of veto players in the budget process is, the more stable the relevant budgetary variables (e.g., total budgetary expenditures and their structure) are.

However, researchers working in the public choice tradition suggest, according to the tragedy of the fiscal commons, that the number of political entities involved in the budgetary process positively influences budgetary expenditures and creates so-called deficit bias [5,8]. Therefore, one could hypothesize that on the one hand institutional constraints on the democratically elected politicians, often guarded by unelected officials should probably favor limiting public expenditures [49], and on the other the sheer number of political actors with different constituencies works in the opposite direction.

There is rich theoretical and empirical literature concerning the consequences of diverse institutional systems affecting the ways of power division between the center and local governments as well as the size and structure of public spending at both higher and lower levels [50–52]. On the other hand, there are forces that act in the opposite direction, encouraging spending to be increased by federation units, in particular in the event of high transfers from the federal government (in connection with the so-called flypaper effect).

### 2.3. Rules of the Budgetary Process

The current of research into the influence of lower-level rules (especially fiscal rules) on budget results is relatively younger than the previously described. There is still a relative lack of research on the endogeneity of these rules [53], and the existing ones point out above all that the actors operating at the constitutional level are aware of the consequences of individual institutions in the context of the tragedy of fiscal commons. Therefore, they more often choose, e.g., rules strengthening the role of the Minister of Finance in the budget process in countries with proportional electoral systems and in countries with a more polarized political scene, where political competition is more visible [54].

As for the direct impact of fiscal rules on budget results, it is difficult to find clear results in the current research into the way the behavior of the actors involved in budget preparation is being shaped by them. For example, Reference [11] indicate that the balanced budget rules and debt rules reduce in principle the interest rate on public debt. But [42] notes that in the event of restrictions on the possibility of issuing debt by US state authorities, they tend to substitute instruments subject to restrictions for those that are regulated more liberally. Reference [55] show the ineffectiveness of the mechanisms introduced by the European Stability and Growth Pact.Reference [56] provide evidence that EU Member States use creative accounting to circumvent the limits of the deficit size. The size of state budget expenditures does not seem to be reduced as a whole as a consequence of introducing restrictions on the possibility of taxing and spending [42,57]. In the event of breaking a fiscal rule, there seems to be a general tendency to subsequent suppression of the fiscal variable being in breach of the fiscal rule towards the benchmark introduced by the rule [58].

### 2.4. Hypotheses

Based on the review of current literature presented above, one can construct several research hypotheses organized around three described categories of institutional operation (i.e., institutions of power legitimacy, of power relations and rules of the budgetary process) and more importantly around the four analyzed theoretical frameworks (tragedy of the fiscal commons, Political Budget Cycle, veto players theory, path dependence).

**Hypothesis 1** (H1). *States with a proportional electoral system are characterized by an average higher level of public spending (institutions of power legitimacy/tragedy of the fiscal commons).*

**Hypothesis 2** (H2). *States with a large size of winning coalition in relation to the size of the electorate, are characterized by a higher level of public spending (institutions of power legitimacy/ tragedy of the fiscal commons).*

**Hypothesis 3** (H3). *There is no visible connection between the time left to the nearest elections and the volume of budget expenditures (institutions of power legitimacy/Political Budget Cycle).*

**Hypothesis 4** (H4)**.** *Governments with a large number of veto players are characterized by higher expenditures (institutions of internal power relations/ tragedy of the fiscal commons).*

**Hypothesis 5** (H5)**.** *Institutional checks and balances introducing independent counteracting forces to the freedom of spending are generally effective (institutions of internal power relations/ veto players theory).*

**Hypothesis 6** (H6)**.** *Fiscal rules, designed to stiffen the budget formation process, in practice have low effectiveness (rules of the budgetary process/ tragedy of the fiscal commons).*

**Hypothesis 7** (H7)**.** *The more robust the political system (the longer it is sustained) the higher the budgetary expenditures (institutions of power legitimacy/ path dependence).*

Due to the specificity of the used methodology (BMA), the list of variables chosen for the research extends beyond those involved in testing the stated hypotheses. The approach chosen by the authors is thus a mixed one, combining confirmatory and exploratory analysis.

## 3. Materials and Methods

The first subsection introduces the data set used in estimations. The second subsection describes BMA estimation structure, along with the used statistics and choices of model and g priors, as well as jointness measures.

### 3.1. Data and Measurement

The independent variables comprise of data on political institutions within three categories described above (institutions of power legitimacy, of power relations, of budgetary process) obtained from different sources as well as several control economic and social variables. Due to a large number of these variables, their detailed description, together with their sources, can be found in Appendix A. However, the most important variables designed to test the hypotheses posed in Section 3.1 are presented in Table 1.

**Table 1.** Variables chosen to test themain hypotheses of the paper.

| Hypothesis | Variable Name | Variable Description |
|---|---|---|
| H1 | *elec_sys* | Electoral systems: 1. Majoritarian; 2. Mixed; 3. Proportional. |
| | *closed_list* | Closed list variable is a dummy taking two values: If voters cannot choose individual candidates and vote for entire lists —1, otherwise—0 |
| | *dist_house* | Mean District Magnitude in House elections. Weighted average of the number of representatives elected in different size districts, if available. If not, the number of seats is divided by the number of districts (if both are known). |
| H2 | *vot_turn* | Voter turnout in last parliamentary election (%) |
| H3 | *yrs_elec* | Years left in current parliamentary term |
| H4 | *gov_frac* | Government Fractionalization Index. The probability that two deputies picked at random from among the government parties will be of different parties (we understand that there are other methods to gauge electoral competition, such as introduced by [59]. However, we decided to use fractionalization as the most often used measure in these kind of studies). |
| | *no_part* | Number of parties in the government |
| H5 | *checks_bal* | The index of checks and balances equals 1, if the legislature is not chosen in competitive elections or in those in which only the executive has real power. For countries that do not meet this criterion (i.e., democratic states), one of the following conditions increases its value: <br><br> 1. The existence of the head of state. <br> 2. Head of state elected in competitive elections. <br> 3. The opposition controls the legislature. |

**Table 1.** *Cont.*

| Hypothesis | Variable Name | Variable Description |
|---|---|---|
| | | Additionally, in presidential systems the value is increased by 1 when: |
| | | 1. There is more than one chamber of parliament, unless the head of state has a majority in the lower house and there is a closed list system |
| | | 2. There are parties recognized as affiliated with the party of the head of state, but with an ideological position similar to the opposition |
| | | In parliamentary systems, however, the value is increased by 1: |
| | | 1. For each party in the ruling coalition, as long as its votes are necessary to keep the majority in the parliament. |
| | | 2. For each party in a ruling coalition whose ideological position is closer to the opposition than to the prime minister's party. |
| | | 3. The value is lowered by 1 when the open list is functioning |
| | *Polcon3* | Political constraints Index III. The index is composed from the following information: the number of independent branches of government with veto power over policy change, counting the executive and the presence of an elective lower and up https://www.overleaf.com/project/5d1da69847b47b3dfc826b40 per house in the legislature (more branches leading to more constraint); the extent of party alignment across branches of government, measured as the extent to which the same party or coalition of parties control each branch (decreasing the level of constraint); and the extent of preference heterogeneity within each legislative branch, measured as legislative fractionalization in the relevant house (increasing constraint for aligned executives, decreasing it for opposed executives). |
| | *Polcon5* | Political constraints Index V. This index follows the same logic as Political Constraints Index III (*Polcon3*) but also includes two additional veto points: the judiciary and sub-federal entities |
| H6 | *er_nat* | Existence of an expenditure rule at a central level. Yes—1, No—0 |
| | *rr_nat* | Existence of a revenue rule at a central level. Yes—1, No—0 |
| | *bbr_nat* | Existence of a balance budget rule at a central level. Yes—1, No—0 |
| | *dr_nat* | Existence of a debt rule at a central level. Yes—1, No—0 |
| | *dr* | Existence of a debt rule at any level of government. Yes—1, No—0 |
| | *bbr* | Existence of a balance budget rule at any level of government. Yes—1, No—0 |
| H7 | *reg_age* | Current political regime durability in years (averages) |

Sources for the data are relegated to Appendix A.

The particular dependent variable used in the analysis (*COFOG_tot*) is the sum of central government expenditures excluding social security contributions by functions of government (COFOG) as a share of GDP compiled by the IMF for the database Government Finance Statistics (COFOG expenditures are divided into the following ten functions: general public services; defense; public order and safety; economic affairs; environmental protection; housing and community amenities; health; recreation, culture and religion; education; and social protection. Data on central government expenditures by function include transfers between the different levels of government). Although it slightly differs from other measures of central government spending, its advantage is completeness for the countries included in the dataset and, due to the same methodology, comparability to data on COFOG expenditures obtained from other sources (e.g., OECD, Eurostat). A variable for budgetary expenditure on the central level has been chosen deliberately. Since most of the institutions we study work at the level of the central government, they may not be well suited for explaining expenditure of local and regional governments. Therefore, the authors have decided to choose a variable which explicitly excludes these expenditures. Such an approach should result in expenditures' levels being more sensitive to changes in institutions. In fact, the variance of spending both between individual countries and in different years is considerable (see Appendix F). This should make it possible to recognize more theories as compatible with each other.

The dataset includes the following 24 countries (members of OECD and Bulgaria) for which we were able to obtain complete data over 2001–2012 (The choice of this particular timespan is informed by

data availability). years for all the variables: Australia, Austria, Belgium, Bulgaria, the Czech Republic, Denmark, Estonia, Finland, France, Germany, Hungary, Ireland, Israel, Italy, Latvia, Luxembourg, the Netherlands, Norway, Poland, Portugal, Slovenia, Spain, Sweden, United Kingdom, and United States. We decided to use this particular sample of countries due to their broadly similar level of development. However, we wanted to use as much data on emerging countries as possible to test, whether the tested theories fare well in somewhat different circumstances than normally assessed. Some countries (such as e.g., Romania) were left out due to lack of data (excluding data on Bulgaria from the dataset does not alter the results in any qualitative manner—results are available upon request). Overall, the panel comprises 300 observations. All 39 variables were tested using first generation tests for the common unit root using [60] and for individual unit root using [61] as well as using second generation test proposed by Pesaran [62]. As confirmed by the aforementioned tests, all variables used in the estimations are weakly stationary (Results are not reported here for brevity and are available upon request). Descriptive statistics and means for all variables in every country are reported in Appendix D.

*3.2. BMA—Bayesian Model Averaging*

The theoretical literature contains a long list of potential institutional determinants of budgetary expenditures. Thus far, researchers were trying to verify the hypotheses about them, focused their inquiries only on a few variables representing the institutions they were interested in or those associated with a given strain of theory. This type of approach completely disregards uncertainty about the specification of the model being tested. This issue is amplified by the presence of open-endedness, the idea that the validity of one casual theory is not implying falsification of another one [63]. With the vast theoretical and empirical literature on the subject, the assessment which of the theories are correct becomes infeasible due to the bulk of inconsistent or even conflicting results that cannot be compared.

Accordingly, in order to assess which institutions are in fact, determining budgetary expenditure, the analytical framework needs to allow the comparison of the different models as well as their assessment based on the empirical grounds. Bayesian model averaging is a method possessing all these qualities, and consequently, in the present paper, it was used to identify the robustness of the most prominent institutional determinants of budget expenditure among the candidates which form the up to date research.

The data comprises a panel of 25 countries over the 2001–2012 period with one dependent variable and 38 regressors. In the literature, country heterogeneity in the data is dealt with using random or fixed effects models. Those models are well fit when a single theory is tested at a time, and random and fixed effects serve as a way of covering up the ignorance about the sources of heterogeneity [64]. On the other hand, BMA deals with heterogeneity directly by finding a combination of regressors which accounts for it to the greatest extent within a conditioning set of information. Consequently, BMA appears to be ideally suited for finding robust determinants of budgetary expenditure. Within the set of regressors, the research strives at the identification of the variables, whose influence on budgetary expenditures finds the most substantial support in the data. BMA assumes the following general form of the model:

$$y = X_j \beta_j + \varepsilon_j \tag{1}$$

where $j = 1, 2, ..., m$ denotes the number of the model, $y_j$ is a vector $((n*t)1)$ of the values of the dependent variable, $\alpha_j$ is a vector of intercepts, $\beta_j$ is a vector $(K \times 1)$ of unknown parameters, $X_j$ is a matrix $((n*t) \times K)$ of explanatory variables, whereas $\varepsilon_j$ is a vector of residuals which are assumed to be normally distributed and conditionally homoscedastic, $\varepsilon \sim N(0, \sigma^2 I)$. $n*t$ denotes the number of observations (300), and $K$ is the total number of regressors (38).

For the space of all models that can be estimated with the 38 regressors at hand, unconditional posterior distribution of coefficient $\beta$ is given by:

$$P(\beta|y) = \sum_{j=1}^{2^K} P(\beta|M_j, y) \times P(M_j|y) \tag{2}$$

where: $y$ denotes data, $j(j = 1, 2, ..., m)$ signify the number of the model, $K$ being the total number of potential regressors, $P(\beta|M_j, y)$ is the conditional distribution of coefficient $\beta$ for a given model $M_j$, and $P(M_j|y)$ is the posterior probability of the model. Using the Bayes' theorem, the posterior probability of the model (PMP—Posterior Model Probability) $P(M_j|y)$ can be rendered as:

$$PMP = P(M_j|y) = \frac{L(y|M_j) \times P(M_j)}{P(y)}, \tag{3}$$

where PMP is proportional to the product of $L(y|M_j)$—model specific marginal likelihood-and $P(M_j)$—model specific prior probability-which can be written down as $P(M_j|y) \propto L(y|M_j) * P(M_j)$. Moreover, because: $P(y) = \sum_{j=1}^{2^K} L(y|M_j) * P(M_j)$, weights of individual models can be transformed into probabilities through the normalization in relation to the space of all $2^K$ models:

$$P(M_j|y) = \frac{L(y|M_j) \times P(M_j)}{\sum_{j=1}^{2^K} L(y|M_j) \times P(M_j)}. \tag{4}$$

Applying BMA requires specifying the prior structure of the model. The value of the coefficients $\beta$ is characterized by normal distribution with zero mean and variance $\sigma^2 V_{0j}$, hence:

$$P(\beta|\sigma^2, M_j) \sim N(0, \sigma^2 V_{0j}). \tag{5}$$

It is assumed that the prior variance matrix $V_{0j}$ is proportional to the covariance in the sample: $(gX_j'X_j)^{-1}$, where $g$ is the proportionality coefficient. The g prior parameter was put forward by [65] and is widely used in BMA applications. In all the estimations presented in this paper two versions of g prior recommended by [66] in their seminal were used, namely: UIP—unit information prior [67], and RIC—risk inflation criterion [68]. For further discussion on the subject of g priors see: [69–72].

While applying BMA, besides the specification of g prior, it is necessary to determine the prior model distribution. In the main results uniform model prior [73] was used, where priors on all the models are all equal ($P(M_j) \propto 1$). Under uniform model prior, the prior probability of including a variable in a model amounts to 0.5. The main estimation results presented in this paper are based on a combination of uniform model prior and unit information g prior. This combination of priors is recommended by [72]. To assure robustness of the results, other prior structures have been used as well. First of all, risk inflation prior advocated by [66] was combined with binomial-beta model prior [69]. In the case of binomial-beta distribution, the probability of a model of each size is the same ($\frac{1}{K+1}$). Thus, the prior probability of including the variable in the model amounts to 0.5, for both binomial and binomial-beta prior with $Em = K/2$. In order to account for potential multicollinearity between regressors, dilution prior was used. Accordingly, a uniform model prior is supplemented with a function accounting for multicollinearity [74] to obtain prior model probabilities:

$$P(M_j) \propto |R_j|^{0.5} \left(\frac{1}{2}\right)^K. \tag{6}$$

where K = (38) is the number of covariates, while $|R_j|$ is the determinant of the correlation matrix for all the regressors in the model j. The uniform model prior implies equal probabilities assigned to all the models, so the $|R_j|$ component of (13) decides about the distribution of the prior probability mass. The higher the multicollinearity between the variables, the closer the value of $|R_j|$ to 0 and

the lower the prior ascribed to a given model. In case of 38 covariates the entire model space consists of around 275 billion possible models, which is a number infeasible to assess analytically. Accordingly, the model space is reduced with $MC^3$ (Markov Chain Monte Carlo model Composition) sampler [75]. The convergence of the chain is assessed by the correlation coefficient between the analytical and MC3 posterior model probabilities for the best 10,000 models. Using the posterior probabilities of the models in the role of weights allows calculation of the unconditional posterior mean and standard deviation of the coefficient $\beta_i$. Posterior mean (PM) of the coefficient $\beta_i$, independent of the space of the models, is then given with the following formula:

$$PM = E(\beta_i|y) = \sum_{j=1}^{2^K} \hat{\beta}_{i,j} \times P(M_j|y) \tag{7}$$

where $\hat{\beta}_{i,j} = E(\beta_i|y, M_j)$ is the value of the coefficient $\beta_i$ estimated for the model $M_j$. The posterior standard deviation (PSD) is equal to:

$$PSD = \sqrt{\sum_{j=1}^{2^K} V(\beta_{k,j}|y, M_j) \times P(M_j|y) + \sum_{j=1}^{2^K} [\beta_{k,j} - E(\beta_{k,j}|y, M_j)]^2 \times P(M_j|y)} \tag{8}$$

where $V(\beta_{i,j}|y, M_j)$ signifies the conditional variance of the parameter for the model $M_j$. To better capture the relative impact of the determinants on the government expenditure, standardized coefficients were calculated and BMA statistics based on their values. SPM denotes the standardized posterior mean, while SPSD denotes a standardized posterior standard deviation (See [76] for elaboration).

$$PIP = P(x_i|data) = \sum_{j=1}^{2^K} 1(x_i = 1|y, M_j) \times PMP_j \tag{9}$$

where $x_i = 1$ signifies including the variable $x_i$ in the model. In uniform and beta-binomial prior model distributions prior inclusion probability is equal to 0.5 and can serve as a point of reference in assessment of the robustness. Following [67,77], the robustness of each regressor is weak, positive, strong, or decisive if the posterior inclusion probability PIP lies between 0.5–0.75, 0.75–0.95, 0.95–0.99, or 0.99–1, respectively. In the case of dilution prior, there is a problem with setting the exact value of prior inclusion probability. As the method combines a uniform model prior with a function penalizing for multicollinearity, the exact prior distribution is not known before calculations. As explained above, the entire model space as well as all the values of $|R_j|$ are infeasible to calculate with a large number of regressors and, consequently, the same is true for prior inclusion probability. On the other hand, $|R_j|$ takes lower values for bigger models by virtue of its construction and, consequently, the expected model size is lower than for uniform distribution, and prior inclusion probability is lower than 0.5 (Moreover, prior inclusion probabilities are lower for the variables characterized by a higher degree of multicollinearity). In this setting, the critical values proposed by [67,77] can serve as a very strict criteria of asserting robustness of the variables under consideration.

Additionally, the researcher can be interested in the sign of the estimated parameter, if it is included in the model. The posterior probability of positive sign of the coefficient in the model $[P(+)]$ is calculated in the following way:

$$\begin{aligned} P(+) &= P[sign(x_i)|y] = \\ &\sum_{j=1}^{2^K} P(M_j|data) \times CDF(t_{k,j}|M_j) \ if \ sign(E(\beta_{k,j}|data, M_j)]^2) = 1 \\ &1 - \sum_{j=1}^{2^K} P(M_j|data) \times CDF(t_{k,j}|M_j) \ if \ sign(E(\beta_{k,j}|data, M_j)]^2) = -1 \end{aligned} \tag{10}$$

where CDF signifies a cumulative distribution function, while $t_{ij} \equiv \hat{\beta}_i / S\hat{D}_i|M_j$.

Within BMA, it is possible to assess the nature of the relationships between regressors using jointness measures. Reference [76] define their jointness measure as:

$$JDW_{ih} = \ln\left[\frac{P(i \cap h|y)}{P(i \cap \bar{h}|y)} * \frac{P(\bar{i} \cap \bar{h}|y)}{P(\bar{i} \cap h|y)}\right] = \ln\left[\frac{P(i|h,y)}{P(\bar{i}|h,y)} * \frac{P(\bar{i}|\bar{h},y)}{P(i|\bar{h},y)}\right] \tag{11}$$

where $i$ and $h$ represent two regressors in the model. One of the biggest drawbacks of JDW is that, by construction, there are circumstances in which it cannot be calculated (For example, when a given variable is characterized by PIP very close to 0 computation of JDW will require division of 0/0, which gives undefined symbol, or nan—not a number). Accordingly, in order to obtain more reliable information about jointness, Reference [78] measure is calculated as:

$$JLS_{ih} = \ln\left[\frac{P(i \cap h|y)}{P(i \cap \bar{h}|y) + P(\bar{i} \cap h|y)}\right] = \ln\left[\frac{P(i \cap h|y)}{P(i|y) + P(h|y) - 2 * P(i \cap h|y)}\right] \tag{12}$$

For both jointness measures ($J$) the same critical values can be applied. When $J > 2$, two variables are referred to as strong complements, $2 > J > 1$ as significant complements, $1 > J > -1$ as unrelated, $-1 > J > -2$ as significant substitutes, while $-2 > J$ signifies strong substitutes [79]. As demonstrated in [80,81], JLS generally outperforms JDW. Accordingly, interpretations of jointness measures in the results are mainly based on JLS (More on jointness measures can be found in [82]).

## 4. Analysis of the Results

The main results of the analysis are presented in Table 2 (Results for combination of RIC and binomial-beta priors, as well as for dilution and UIP prior are shown in Appendix B. Jointness measure are relegated to Appendix C). Appendix E shows graphically the inclusion probability of every variable in 1000 best models. Besides that, the section is organized as follows. First the results for the three institutional levels (of power legitimacy, of internal relations and rules of the budgetary process) are presented subsequently. Then an analysis of supplementary and control variables is conducted to explore avenues of further research. The section is concluded with a part describing the most important predictors of budgetary expenditures.

**Table 2.** Main results of the analysis.

| Model Prior | Uniform | | | | | Dilution | | | | |
|---|---|---|---|---|---|---|---|---|---|---|
| g Prior | Unit Information Prior | | | | | Risk Inflation Criterion | | | | |
| VARIABLE | PIP | PM | PSD | SPM | SPSD | P(+) | PIP | PM | PSD | SPM | SPSD | P(+) |
| bbr_nat | 1.000 | 4.051 | 0.596 | 0.293 | 0.043 | 1.000 | 1.000 | 4.098 | 0.598 | 0.297 | 0.043 | 1.000 |
| English_LE | 1.000 | 11.710 | 1.220 | 0.679 | 0.071 | 1.000 | 1.000 | 11.340 | 1.009 | 0.657 | 0.058 | 1.000 |
| French_LE | 1.000 | 12.660 | 1.564 | 0.824 | 0.102 | 1.000 | 1.000 | 12.901 | 1.440 | 0.839 | 0.094 | 1.000 |
| Socialist_LE | 1.000 | 11.790 | 1.546 | 0.767 | 0.101 | 1.000 | 1.000 | 11.198 | 1.506 | 0.729 | 0.098 | 1.000 |
| elec_sys | 1.000 | 4.119 | 0.601 | 0.449 | 0.065 | 1.000 | 1.000 | 4.145 | 0.629 | 0.451 | 0.068 | 1.000 |
| cur_union | 1.000 | −8.231 | 0.772 | −0.596 | 0.056 | 0.000 | 1.000 | −8.415 | 0.769 | −0.609 | 0.056 | 0.000 |
| reg_age | 1.000 | 0.073 | 0.009 | 0.514 | 0.065 | 1.000 | 1.000 | 0.073 | 0.009 | 0.514 | 0.064 | 1.000 |
| checks_bal | 1.000 | −1.785 | 0.220 | −0.284 | 0.035 | 0.000 | 1.000 | −1.816 | 0.219 | −0.289 | 0.035 | 0.000 |
| closed_list | 1.000 | −5.998 | 0.806 | −0.434 | 0.058 | 0.000 | 1.000 | −6.217 | 0.826 | −0.450 | 0.060 | 0.000 |
| advanced | 1.000 | 5.371 | 0.831 | 0.285 | 0.044 | 1.000 | 1.000 | 5.224 | 0.824 | 0.278 | 0.044 | 1.000 |
| resour_rich | 1.000 | 9.931 | 1.424 | 0.282 | 0.040 | 1.000 | 1.000 | 10.051 | 1.384 | 0.285 | 0.039 | 1.000 |
| religion_frac | 1.000 | −10.005 | 1.557 | −0.319 | 0.050 | 0.000 | 1.000 | −10.453 | 1.430 | −0.334 | 0.046 | 0.000 |
| German_LE | 1.000 | 14.889 | 1.672 | 0.585 | 0.066 | 1.000 | 1.000 | 14.604 | 1.555 | 0.574 | 0.061 | 1.000 |
| fed | 1.000 | −5.424 | 1.154 | −0.314 | 0.067 | 0.000 | 0.993 | −4.627 | 1.277 | −0.268 | 0.074 | 0.000 |
| ethnic_frac | 1.000 | −17.659 | 2.688 | −0.458 | 0.070 | 0.000 | 1.000 | −19.496 | 1.805 | −0.505 | 0.047 | 0.000 |
| polcon5 | 1.000 | −15.661 | 3.269 | −0.168 | 0.035 | 0.000 | 1.000 | −16.384 | 3.133 | −0.176 | 0.034 | 0.000 |
| er_nat | 1.000 | −2.475 | 0.498 | −0.173 | 0.035 | 0.000 | 1.000 | −2.663 | 0.472 | −0.186 | 0.033 | 0.000 |

**Table 2.** *Cont.*

| Model Prior | Uniform | | | | | | Dilution | | | | | |
|---|---|---|---|---|---|---|---|---|---|---|---|---|
| g Prior | Unit Information Prior | | | | | | Risk Inflation Criterion | | | | | |
| VARIABLE | PIP | PM | PSD | SPM | SPSD | P(+) | PIP | PM | PSD | SPM | SPSD | P(+) |
| *gov_frac* | 0.997 | 9.846 | 1.964 | 0.364 | 0.073 | 1.000 | 0.992 | 10.371 | 1.558 | 0.383 | 0.058 | 1.000 |
| *pub_bal* | 0.935 | −0.576 | 0.176 | −0.413 | 0.126 | 0.000 | 0.931 | −0.599 | 0.173 | −0.429 | 0.124 | 0.000 |
| *vot_turn* | 0.875 | 0.063 | 0.033 | 0.111 | 0.059 | 1.000 | 0.492 | 0.032 | 0.036 | 0.057 | 0.064 | 1.000 |
| *dr_nat* | 0.813 | −1.349 | 0.822 | −0.082 | 0.050 | 0.000 | 0.764 | −1.434 | 0.918 | −0.087 | 0.056 | 0.000 |
| *debt_pub* | 0.592 | 0.017 | 0.017 | 0.071 | 0.070 | 1.000 | 0.336 | 0.011 | 0.017 | 0.045 | 0.068 | 1.000 |
| *language_frac* | 0.276 | −1.724 | 3.321 | −0.049 | 0.094 | 0.014 | 0.015 | −0.096 | 0.879 | −0.003 | 0.025 | 0.008 |
| *polcon3* | 0.263 | 0.998 | 1.954 | 0.017 | 0.034 | 1.000 | 0.062 | 0.231 | 1.023 | 0.004 | 0.018 | 1.000 |
| *no_part* | 0.164 | 0.097 | 0.274 | 0.017 | 0.049 | 1.000 | 0.024 | 0.027 | 0.197 | 0.005 | 0.035 | 1.000 |
| *bud_bal* | 0.146 | −0.058 | 0.177 | −0.040 | 0.121 | 0.000 | 0.070 | −0.047 | 0.174 | −0.032 | 0.119 | 0.000 |
| *gdpgr* | 0.124 | −0.010 | 0.034 | −0.005 | 0.017 | 0.000 | 0.044 | −0.003 | 0.020 | −0.002 | 0.010 | 0.000 |
| *yrs_elec* | 0.111 | −0.018 | 0.068 | −0.003 | 0.012 | 0.000 | 0.048 | −0.008 | 0.045 | −0.001 | 0.008 | 0.000 |
| *dist_house* | 0.109 | −0.002 | 0.008 | −0.006 | 0.027 | 0.135 | 0.010 | 0.000 | 0.002 | 0.000 | 0.007 | 0.023 |
| *unemployment* | 0.106 | −0.009 | 0.038 | −0.005 | 0.020 | 0.014 | 0.023 | −0.002 | 0.019 | −0.001 | 0.010 | 0.006 |
| *rr_nat* | 0.090 | 0.074 | 0.344 | 0.004 | 0.017 | 1.000 | 0.014 | 0.008 | 0.122 | 0.000 | 0.006 | 0.994 |
| *dr* | 0.086 | 0.056 | 0.291 | 0.003 | 0.016 | 0.978 | 0.016 | 0.012 | 0.132 | 0.001 | 0.007 | 0.995 |
| *x2009* | 0.075 | −0.041 | 0.257 | −0.002 | 0.010 | 0.011 | 0.025 | −0.009 | 0.123 | 0.000 | 0.005 | 0.013 |
| *e_union* | 0.061 | −0.025 | 0.272 | −0.001 | 0.015 | 0.267 | 0.006 | 0.001 | 0.076 | 0.000 | 0.004 | 0.461 |
| *elec_year* | 0.061 | 0.009 | 0.096 | 0.001 | 0.006 | 0.907 | 0.027 | 0.004 | 0.062 | 0.000 | 0.004 | 0.971 |
| *bbr* | 0.060 | 0.014 | 0.212 | 0.001 | 0.009 | 0.892 | 0.012 | 0.004 | 0.092 | 0.000 | 0.004 | 0.984 |
| *inflation* | 0.057 | 0.001 | 0.026 | 0.000 | 0.007 | 0.795 | 0.020 | 0.001 | 0.015 | 0.000 | 0.004 | 0.973 |
| *x2010* | 0.056 | −0.009 | 0.146 | 0.000 | 0.006 | 0.010 | 0.023 | −0.004 | 0.093 | 0.000 | 0.004 | 0.007 |
| Burn-ins | 100,000 | | | | | | | | | | | |
| Iterations | 1 m | | | | | | | | | | | |
| Cor PMP | 0.9987 | | | | | | 0.9998 | | | | | |

*4.1. Institutions of Power Legitimacy*

The nominal variable (*elec_sys*) responsible for the electoral system (majoritarian, mixed or proportional) is robust with PIP equal to 1, and PM 4.12. In other words, in line with established research to date in the vein of the "tragedy of the fiscal commons" tradition, the more proportional the electoral system, the higher government expenditures [22,23].

Conversely a dummy variable for closed list (*closed_list*) is characterized by PIP equal to 1 and posterior mean −6. In other words, the existence of closed list in a particular political regime is associated with 6 percentage points lower government expenditures on average. This lends credibility to the view that closed electoral lists cause more party discipline outside majoritarian systems, so it is easier to control spending by party leadership in a top-down manner, without taking into account the preferences of all candidates [34].

Voter turnout in parliamentary elections (*vot_turn*) was classified as fragile, but with 0.88 posterior inclusion probability we choose to interpret its impact, bearing in mind that it does not reach the high threshold of PIP equal to 0.9. PM is equal to 0.06 indicating that ten percentage points higher turnout can be related to 6 per mil points increase in expenditures, which can be explained in the vein of selectorate theory, by the decision to increase the burden on the system for the provision of general public goods and increasing fiscal transfers. This is, however, inconsistent with some other studies which find negative [83,84] or insignificant influence [85]. We favor the explanation that increased electoral participation can be associated with amplified pressure on politicians to generate more inclusive budgets, especially towards lower income population strata, which is consistent with some other research [36]. Attesting to some mentioned disagreement in the literature, the variable appears fragile in all specifications.

Interestingly, district magnitude (*dist_house*) does not seem to be a strong determinant of government expenditure either. This may explain a disagreement in the literature on the relation between the variables [23,24]. The *yrs_elec* and *elec_year* also obtain low PIPs (0.111 and 0.061) which suggest that the political budget cycle in advanced economies is at most weak, confirming established results for developed countries [17,28]. As a rule, older democracies are characterized by higher government expenditure. The PIP for the variable *reg_age* equals 1, while posterior mean suggests that one additional year of democracy is connected with 0.07 percentage points increase in expenditures. This is consistent with the path dependence hypotheses posed by e.g., [1,86,87], and others which state that the longer period of regime stability one faces the more resources are redirected through public sphere. This also suggests that in the absence of forces opposing the expansion of public spending, democratic processes are naturally biased in favor of replacing private sector activities with public action in the long term [3] and corresponds well with the Peacock-Wiseman hypothesis of long-run changes in the intensity of state activity in response to short-run disturbances such as crises, natural disasters and wars [88].

Government Fractionalization (*gov_frac*) as the approximation of the weighted number of political veto players (parties) in the central government is robust, and in line with the prediction resulting from the concept of the tragedy of the fiscal commons increases government expenditures [89–91], as indicated by PM of 9.85. Moreover, the variable strongly complements the impact of electoral system on expenditures (JLS eq. 5.67), suggesting that government fractionalization is more of a problem for proportional systems than majoritarian ones. A similar direction of impact is assumed for the variable *no_part* (number of parties in the government), i.e., the unweighted index of government fractionalization, although the variable is characterized by PIP far below the threshold for robust variables. Additionally, since in line with the Duverger's law electoral system is a strong predictor of both government fractionalization and the number of parties, it is sensible to look for jointness measures. Government fractionalization is a strong complement of electoral system variable with JLS measure of 5.67. This means that more proportional systems are more affected by government fractionalization in terms of the influence on central government expenditures. Conversely, the number of parties is a significant substitute of electoral system variable (JLS eq. -1,63). Together this may mean that the systems which generate governments with a large number of equally strong parties in the coalition (high fractionalization index) suffer most suffer the highest increase in expenditures.

Also, lending credibility to the concept of the tragedy of the fiscal commons, all institutional restrictions imposed on actors in the public sphere in the form exemplified by the highly complementary *Polcon5* and *checks_bal* variables negatively affect the ability to generate high government expenditures [49]. *Polcon5* is characterized by PIP of 1, while the posterior probability of a positive sign is 0. The same situation occurs with *checks_bal* variable.

*Polcon3* variable is below the threshold, but the sign is positive (contrary to *Polcon5*). In this context, one should observe that *Polcon3* is more biased towards partisan veto points just like government fractionalization and the number of parties), while *Polcon5* includes more institutional veto points (federal structure and independent judiciary), which makes it more similar to *checks_bal* variable. This makes for an observation that in a system with more partisan-democratic veto points we can witness a pressure towards increasing expenditure, while with more actors shielded from the electoral assessment the opposite is the case.

## 4.2. Rules of the Budgetary Process

An expenditure rule introduced at the national level (*er_nat*), in accordance with its function, clearly reduces the amount of government expenditures (on average by 2.5 percentage points, with PIP of 1). The debt rule at the national level (*dr_nat*) has a negative (but according to our specification fragile) impact on the amount of expenditures (PIP of 0.81 and PM of −1.35), which may be a natural consequence of the positive impact of the size of public debt on the size of government expenditures. The balanced budget rule at the national level (*bbr_nat*), in turn, positively influences the amount of

expenditures (PIP of 1 and PM of 4.05), which is puzzling. If this rule is effective, and there are reasons to believe this is the case, it should be able to reduce the deficit and, consequently, public expenditure. Overall, this interesting result may suggest that when balanced budget rule is introduced in countries which experience large deficits and large expenditures simultaneously, they try to narrow the deficit by rising taxes instead of reducing expenditures. Moreover, the strong complementarity of *bbr_nat* and *pub_bal* variables (JLS of 2.8) is worth noting, which suggests in turn that countries which violate their commitment to balance the budget by running large deficits to finance increased expenditure are penalized by more stringent interest rate which increases expenditure even further.

### 4.3. Other Institutional Variables and Controls

As the BMA framework allows for testing many variables concurrently, authors have decided, based on an analysis of research to date, to include variables traditionally deemed important for the fiscal policy in the literature.

Federal states distribute significantly less public resources at the central level, which results both from the basic features of such a form of the decentralization of public activities and from the specificity of this kind of institutional arrangement which favors limited expenditures [50–52]. This notion is confirmed in the results, with the fed variable being robust with posterior mean of $-5.42$. Simply put government expenditures in federal states are on average 5.4 percentage points lower than in unitary states.

As a control, dummy variables for German, French, English, and Socialist legal origins were included in the analysis. Interestingly, all these variables assume PIP of 1 and, consistently, have positive signs with a posterior mean ranging from 11.71 for English legal origin to 14.89 for German legal origins. Conversely, this means that the countries of the Scandinavian legal origins have smaller expenditures than other developed countries. This is a very interesting result that demands deeper explanation. Traditionally, Scandinavian countries (Finland, Denmark, Norway, and Sweden in our database), have been associated with higher levels of public (and government) expenditures due to their emphasis on equality and social spending [92]. This approach to public intervention has been dubbed the "Nordic model" [93]. When we look at the data gathered for discussed research at "face value" we may reach similar conclusions. The average government expenditures in Scandinavian countries in our database are 30.39% (relative to GDP), compared to 25.88% in the rest of the countries. This significant difference, similar to other research to date does not take into account the mediating effect of political institutions on public spending. Based on our results, one can hypothesize that the specificity of Scandinavian countries has more to do with their institutional landscape than with significantly higher nominal spending. Scandinavian countries all use proportional representation systems, have traditionally highest electoral turnout, highest government fractionalization (and the number of parties in the government) and a significantly lower index of institutional constraints (*Polcon5*) than other developed countries and are relatively old democracies. This interesting result makes it possible to argue that the exceptionalism of Scandinavian countries is basically defined by the political institutions that they tend to choose compared to other developed democracies. The negative impact of the "Scandinavian legal origins" seems to be driven by other, possibly cultural factors. This seems to be an interesting avenue of further research.

Similarly, highly developed (*advanced*) countries and Norway as a resource-rich state are characterized by a higher level of government expenditure. PIP and P(+) in case of both dummy variables (*advanced*, *res_rich*) is 1. In the first case, this situation can be again explained by Wagner's law or by the increased capacity to mobilize resources (in the form of taxes and loans) by such countries. The second case is consistent with explanations indicating a stronger position of the public sphere in countries rich in natural resources through their total or partial monopolization [94]. In our sample, only Norway is classified as resource rich, which is a special case due to its very prudent management of oil windfalls, which, however, does not preclude some "expenditure bias" among its politicians aware of the fact that a large sovereign fund is waiting to be tapped into. The PM of the variable is

9.93 which means that after controlling for other institutional characteristics Norway government expenditure is 9.93 percentage points higher than in other countries. This goes some way towards explaining the unexpected (implied) negative impact of the Scandinavian legal origins on expenditures.

Countries of the Eurozone, after taking into account all other institutional conditions, are consistently characterized by lower public spending than other countries (EU and outside the EU). Currency union dummy variable (*cur_union*) has PIP of 1 and posterior mean of −8.23, which means that entering the Eurozone is associated with reduction of expenditure by more than 8 percentage points on average (However, and this comment refers to most of the independent variables, especially dummies, one has to bear in mind that there are probably shared institutional characteristics that distinguish Eurozone members from other countries. In other words there is a potential problem of endogeneity, where other variables may cause e.g., both higher expenditures and higher probability of country being (entering the currency union). This is a risk not controlled for in the modelling strategy chosen here). This lends itself as a justification for the effectiveness of public finance management instruments of the Stability and Growth Pact or the auto-selection of countries entering the Eurozone, which are forced to meet the Maastricht criteria. In the latter case, the influence of the path dependence mechanism can be expected [95]. The difference between expenditures of countries in the Eurozone and outside it are pronounced even after controlling for the effects of the 2008 crisis (see Table 3), after which the former increased spending more visibly than the latter. This does not mean that in general the rules of the Pact do not exhibit problems in other economic and fiscal dimensions, taking into consideration the problems of assymetric Union [96,97]. It is just a statement of their relative effectiveness in terms of central budget expenditures.

**Table 3.** Government expenditure relative to GDP of countries in the currency union (CU - i.e., Eurozone) and outside it (NON-CU) before and after the crisis.

|  | NON_CU | CU |
|---|---|---|
| 2001–2007 | 30.37% | 26.43% |
| 2008–2012 | 31.57% | 29.39% |
| Change | 3.96% | 11.18% |

In accordance with the results of previous research, both ethnic and religious fractionalization negatively affects the amount of expenditures (PIP and P(+) in case of both variables *ethnic_frac* and *religious_frac* equals 1). This can be a result both of the lack of consensus on the direction of expenditure in society (especially limiting expenditure on the needs of minorities-see [98,99]) and from the interaction between social fragmentation and the probability of adopting the federal regime implying a positive relationship (Anderson, 2013). In our sample, federal states are characterized by higher indexes of fractionalization. in addition, both indicators of social fractionalization are strong compliments to the federal variable (JLS equals 8.7 and 9.2 for ethnic and religious fractionalization respectively).

Interestingly, countries with higher public debt seem not to suffer much in terms of bigger expenditure as attested by the fragility of the *debt_pub* variable. In the context of the growing consensus that high debt reduces social spending [100], we may assume that increased spending on debt servicing in the higher-debt countries is mitigated by the reduction in social spending. Overall, this result encourages a more thorough analysis, especially that the relationship proves fragile also in the dilut-RIC and dilut-UIP specifications, which means that it is possible that the public debt does not influence central government expenditure, which could mean that the interest rate and social spending channels cancel each other out.

In the case of public fiscal balance (and budget balance, although here the dependence is weaker and fragile), we observe an intuitive relationship–the higher the public deficit (lower the balance), the higher the government expenditure. PIP of *pub_bal* variable is 0.94 and PM equals −0.58. The explanation for this direction of influence is that deficit financing of expenditures may be more desirable for politicians than non-repayable funding; therefore, as a consequence of possibility of

deficit financing, expenditures are raised to a more extent than when it is possible to finance them only with tax-related instruments. This lends credibility to the fiscal myopia argument, where politicians, together with citizens, discount future obligations highly enough to increase expenditure in the face of increased deficit [101].

Other variables introduced into the model proved to be fragile with PIP for every one of them lower than 0.28. The PIP of language fractionalization (*language_frac*) variable also puts it below the threshold, but the P(+) of 0.014 suggests it as a negative correlate of government expenditure similarly to ethnic and religious fractionalization. It is worth noting that the variable is significantly correlated with ethnic fractionalization (with Pearson r of 0.78) and can be treated as a substitute for it with JLS = −0.95. Budgetary deficit seems to be "insignificant" in the highest weighted models, but the *bud_bal* retains the same sign as *pub_bal*. *Gdpgr* variable is also not included, which suggests that the relationship is ambiguous at most, and probably needs a more sophisticated analysis, comparing economic growth to various components of public expenditure [102]. Unemployment and inflation, as well as revenue rule at the national level, debt rule, balanced budget rule, membership of EU do not seem to affect government expenditures in perceivable way either, once other institutional features are taken into account. Interestingly, dummies for 2009 and 2010 do not seem to be robust determinants of expenditures as well, despite an increase in expenditures in all countries, which may mean that institutional features of particular countries robustly determine their responses to the financial crisis. Detailed analysis of the data suggests that oversized response of public expenditure to the crisis may be part of explanation of positive relationship between *English_LE* and *COFOG_tot* (average expenditure in these countries rose in the period under consideration by 18%, while in the rest it increased by 8%).

## 4.4. Strength of Influence and Robustness

To assess relative strength of influence of examined determinants of growth standardized posterior mean and posterior standard deviation were calculated. The analysis of standardized posterior means for the variables with PIP above the threshold attests to a very strong influence of particular legal origins on government expenditures (SPM between 0.59 to 0.82). Ethnic and linguistic fractionalization also strongly affect expenditures with SPM of −0.46 and −0.32, respectively. Established institutional features such as being a member of currency union, stability of the political regime, electoral systems and being a federation moderately influence expenditures, similarly to government fractionalization and fiscal balance. Some country features which operationalize power relations such as checks and balances and *Polcon5* seem to be weaker determinants of expenditures, similarly to fiscal rules. Relatively weak SPM for advanced variable suggests that there is a visible variability among the developed countries in terms of their public expenditure bias. Voter turnout does not seem to be influencing expenditure significantly, which, in connection with traditional political economy arguments, may suggest that the position of the median voter does change, but not significantly, when preferences of a bigger number of voters are taken into account [103]. This may also be considered in light of the argument put by [5] that political actors are only loosely and conditionally constrained by the will of their constituencies. A very low SPM by the *debt_pub* variable seems to be connected with the complex nature of the influence of public debt on expenditures which operates on at least two levels: as the matter of pure accounting debt levels increase expenditure, while they may compel politicians to somewhat reduce current spending (especially on social programs) in order to minimize the risk of default.

Qualitatively and quantitatively, similar results were obtained under a different prior structure i.e., a combination of unit information g prior and uniform model prior. With the combination of risk inflation criterion g prior and binomial-beta model prior, the same variables were classified as robust, and posterior means were of basically the same values as in the main results. Similar results were obtained with the dilution prior, which accounts for the presence of multicollinearity in the conditioning set of information. Under the combination with, dilution prior, unit information prior

and with risk inflation criterion g prior, results were also qualitatively similar. Consequently, all changes in prior structure show that the main results are very robust.

Additionally, we made two robustness checks of the results. First, we added a proxy for the impact of globalization–we used trade openness as a proxy: exports and imports as a share of GDP. The variable turned out fragile with PIP equal to 0.123. Secondly, we eliminated Bulgaria from the sample. In both instances the results turned out qualitatively unchanged (the results of these additional robustness checks are available upon request).

## 5. Conclusions

The paper, based on a mix of confirmatory and exploratory analysis, allowed to test the hypotheses put forth in Section 3.1 and made an attempt to discriminate among the theories presented in the introduction. Table 4 summarizes the results in terms of the posed hypotheses.

**Table 4.** The results of hypotheses testing.

| Hypothesis | Verdict |
|---|---|
| **H1.** *States with a proportional electoral system are characterized by an average higher level of public spending (institutions of power legitimacy).* | Not rejected |
| **H2.** *States with a large size of winning coalition in relation to the size of the selectorate, are characterized by a higher level of public spending (institutions of power legitimacy).* | Not rejected, but based on the value of PIP and SPM the link is very weak. |
| **H3.** *There is no visible connection between the time left to the nearest elections and the volume of budget expenditures (institutions of power legitimacy).* | Not rejected |
| **H4.** *Governments with a large number of veto players are characterized by higher expenditures (institutions of internal power relations).* | Not rejected |
| **H5.** *Institutional checks and balances introducing independent counteracting forces to the institutions limiting government freedom of spending are generally effective (institutions of internal power relations).* | Not rejected, but based on the value of SPM the link is not very strong. |
| **H6.** *Fiscal rules, designed to stiffen the budget formation process, in practice have low effectiveness (rules of the budgetary process).* | Analysis points towards rejection of H6. expenditure rule at the national level seems to suppress budget expenditures, but the influence of the rule, based on the value of SPM remains limited. |
| **H7.** *The more robust the political system (the longer it is sustained) the higher the budgetary expenditures (institutions of power legitimacy/path dependence).* | Not rejected |

Overall, the presented results unambiguously confirm the well-established strain of research on the tragedy of fiscal commons, both in its first approximation (the number of actors with the power to influence expenditures positively affects them) and its institutional correlates (institutions which are designed to limit political power of the elected politicians work as intended by reducing the expenditure bias of democratic systems). At a more detailed level, it seems that political systems, together with some very fundamental institutional/cultural features (such as legal origins), are the most robust and strong determinants of central government expenditures, followed by institutions structuring power relations and fiscal rules.

The old path dependence arguments suggesting that democracies incrementally increase their expenditures on the development path are also positively verified, with older democracies and richer countries characterized with increased government expenditure. From the perspective of veto player theories, the analysis suggests that in terms of public expenditure, explanations based on this tradition should not avoid nuance as to the type of veto player being analyzed. Based on the results, one can argue that institutional veto points which are weakly bound by the will of the voters decrease (and probably stabilize) public expenditures, while adding veto players of partisan nature increases expenditures. This is yet another analysis which suggests weak explanatory power of the political business/budget cycle theories, at least among developed countries.

Overall, the analysis suggests that in terms of factors influencing government expenditures institutions matter, and they matter significantly. Many of the standard control variables (unemployment, inflation, dummies for post-crisis years, GDP growth) have proven to be unsatisfactory in explaining the levels of expenditure when institutional features of OECD countries are accounted for. Table 5 synthesizes the main conclusions of the paper with respect to four frameworks (or theoretical families).

**Table 5.** Synthesis of the main conclusions of the paper.

| Framework | Fundamental Claim | Evaluation |
|---|---|---|
| Tragedy of the fiscal commons | The more actors with differing political bases are engaged in the budgetary process, the higher the public expenditures (and deficit) are | This claim is positively verified |
| Path dependence | Public expenditures rise incrementally in the long run | This claim is positively verified |
| Veto Players | The more veto players with different ideologies are engaged in the budgetary process, the more difficult it is to change expenditures | Institutional veto players which are weakly bound by the will of the voters decrease (and probably stabilize) public expenditures, while adding veto players of partisan nature increases expenditures. |
| Political Budget Cycle | Expenditures (and deficits) rise before important political elections | This claim is verified negatively |

In terms of suggestions for economic policy one must acknowledge that different institutions have a different impact on public spending. Interestingly, the currently functioning fiscal rules, when taking into account political institutions, except expenditure rules and balanced budget rules on the national level, do not seem to have an overwhelming impact on the volume of government spending. This may mean that they are too easy to bypass by politicians, and are not yet rooted in politics (as witnessed during COVID-19 induced crisis). Perhaps in the long run "fiscal councils" will prove to be useful, however, provided that they are equipped with relatively broad powers and competences.

Political institutions seem to have an indirect but clear impact on the shape and results of the budget process. This statement is quite strong, but at the same time not very useful–no one changes political institutions to deal with the problem of the size of the public sphere. It is simply too expensive a change. In addition, the possibility of changing such institutions, during the maturing and stabilization of social systems, appears less and less often in the form of so-called "windows of opportunity". Systemic changes, although they provide a good opportunity for a radical transformation of institutions, are associated with many side effects that can be detrimental to the societies affected. In addition, these types of changes are usually correlated with fundamental changes at the level of ideas, to which the institutions described in the article may not be adapted.

In conclusion, the authors would like to discuss three potential doubts as to the particular methodological choices made in the article. First, it may be argued that general government expenditure constitutes a better dependent variable in this kind of analysis compared to central government expenditure because it is a better proxy for the activities within the whole public sector. The authors acknowledge the fact that a large part of activities in the public sector is happening below the central government level. However, in decentralized countries, the relationship between political institutions designed as constraints on the central government and overall expenditure is much more complex, and needs to consider a lot more variables and interdependencies between them. We plan to undertake this problem in future investigations. Secondly, country heterogeneity in the data is traditionally dealt with using random or fixed effects models, unlike in this paper, where the problem was resolved with the usage of BMA. However, as already argued in the methodological section, traditional models are well fit when one given theory is tested at a time, and random and fixed effects serve as a way of covering up the ignorance about the sources of heterogeneity [64]. From this point

of view the study aims at overcoming the problem of the lack of prior knowledge with the help of methodology dealing with heterogeneity on empirical grounds. Thirdly we do not encompass all important determinants of public spending—such as e.g., political ideology of political incumbents. This was a deliberate choice informed by the decision to focus on the institutional dimension of political systems.

**Author Contributions:** The authors contributed equally to the research presented in the paper and the preparation of the final manuscript. All authors have read and agreed to the published version of the manuscript.

**Funding:** The publication is financed by the Ministry of Science and Higher Education of Poland within "Regional Initiative of Excellence" Programme for 2019–2022. Project no.: 021/RID/2018/19. Total financing: 11 897 131,40 PLN. The APC was funded by the same project.

**Conflicts of Interest:** The authors declare no conflict of interest.

## Appendix A. Description of All Variables Employed in Estimations

**Table A1.** List of variables with explanation.

| Short Name | Explanation | Source |
|---|---|---|
| *er_nat* | Explained in Table 1 in the main text | [104] (IMF Fiscal rules dataset) |
| *rr_nat* | Explained in Table 1 in the main text | |
| *bbr_nat* | Explained in Table 1 in the main text | |
| *dr_nat* | Explained in Table 1 in the main text | |
| *dr* | Explained in Table 1 in the main text | |
| *bbr* | Explained in Table 1 in the main text | |
| *cur_union* | Dummy answering the question: Is country a member of currency union? 1—yes, 0—no | |
| *advanced* | Dummy answering the question: Country is on IMF the list of Advanced Economies—1, no—0 | |
| *resour_rich* | Dummy answering the question: Is the economy rich in natural resources? 1—yes, 0—no | |
| *fed* | Dummy answering the question: Is country a federation—1, otherwise—0 | |
| *e_union* | Dummy for EU members Yes—1, No—0 | |
| *English_LE* | According to "legal origins" concept one of possible legal systems: English Common Law | [105] |
| *French_LE* | According to "legal origins" concept: French Commercial Code, | |
| *Socialist_LE* | According to "legal origins" concept: Socialist/Communist Laws, | |
| *German_LE* | According to "legal origins" concept: German Commercial Code, | |
| *elec_sys* | Explained in Table 1 in the main text | [106] |
| *reg_age* | Current political regime durability in years (averages) | Polity IV project |
| *checks_bal* | Explained in Table 1 in the main text | [107] |
| *closed_list* | Explained in Table 1 in the main text | |
| *gov_frac* | Explained in Table 1 in the main text | |
| *no_part* | Explained in Table 1 in the main text | |
| *yrs_elec* | Explained in Table 1 in the main text | |
| *dist_house* | Explained in Table 1 in the main text | |
| *elec_year* | Dummy for parliamentary election year. Yes—1, No—0 | |

**Table A1.** *Cont.*

| Short Name | Explanation | Source |
|---|---|---|
| *ethnic_frac* | Ethnic fractionalization. The variable reflect the probability that two randomly selected people from a given country will not share ethnicity, the higher the number the less probability of the two sharing that characteristic. | [108] |
| *language_frac* | Linguistic fractionalization. The variable reflect the probability that two randomly selected people from a given country will not share language, the higher the number the less probability of the two sharing that characteristic. | |
| *religion_frac* | Religious fractionalization. The variable reflect the probability that two randomly selected people from a given country will not share religion, the higher the number the less probability of the two sharing that characteristic. | |
| *polcon3* | Explained in Table 1 in the main text | [109] |
| *polcon5* | Explained in Table 1 in the main text | |
| *vot_turn* | Explained in Table 1 in the main text | [110] |
| *pub_bal* | Public fiscal net balance. Surplus (+)/Deficit (-) | IMF |
| *bud_bal* | Budgetary balance. Surplus (+)/Deficit (-) | |
| *debt_pub* | Gross General Government Debt % GDP | |
| *gdpgr* | GDP growth (%) | |
| *unemployment* | Official unemployment rate (%) | |
| *inflation* | Control for the inflation rate (%) | |
| *COFOG_tot* | Total central government expenditures (%GDP) - dependent variable | |
| *x2009* | Control for first year after the financial crisis | Own |
| *x2010* | Control for second year after the financial crisis | |

## Appendix B. Result for the RIC and Binomial-Beta, as Well as UIP and Dilution Prior

**Table A2.** Results for the RIC and binomial-beta, as well as UIP and dilution prior.

| Model Prior | Beta-Binomial | | | | | | Dilution | | | | | |
|---|---|---|---|---|---|---|---|---|---|---|---|---|
| g Prior | Risk Inflation Criterion | | | | | | Unit Information Prior | | | | | |
| Variable | PIP | PM | PSD | SPM | SPSD | P(+) | PIP | PM | PSD | SPM | SPSD | P(+) |
| er_nat | 1.000 | −2.519 | 0.493 | −0.176 | 0.034 | 0.000 | 1.000 | −2.62 | 0.477 | −0.183 | 0.033 | 0.000 |
| bbr_nat | 1.000 | 4.054 | 0.593 | 0.294 | 0.043 | 1.000 | 1.000 | 4.052 | 0.604 | 0.293 | 0.044 | 1.000 |
| English_LE | 1.000 | 11.639 | 1.187 | 0.675 | 0.069 | 1.000 | 1.000 | 11.372 | 1.044 | 0.659 | 0.061 | 1.000 |
| French_LE | 1.000 | 12.685 | 1.534 | 0.825 | 0.100 | 1.000 | 1.000 | 12.623 | 1.456 | 0.821 | 0.095 | 1.000 |
| Socialist_LE | 1.000 | 11.695 | 1.544 | 0.761 | 0.100 | 1.000 | 1.000 | 11.373 | 1.52 | 0.74 | 0.099 | 1.000 |
| elec_sys | 1.000 | 4.125 | 0.607 | 0.449 | 0.066 | 1.000 | 1.000 | 4.065 | 0.629 | 0.443 | 0.069 | 1.000 |
| cur_union | 1.000 | −8.26 | 0.766 | −0.598 | 0.055 | 0.000 | 1.000 | −8.264 | 0.772 | −0.598 | 0.056 | 0.000 |
| reg_age | 1.000 | 0.073 | 0.009 | 0.515 | 0.065 | 1.000 | 1.000 | 0.073 | 0.009 | 0.515 | 0.065 | 1.000 |
| checks_bal | 1.000 | −1.793 | 0.219 | −0.286 | 0.035 | 0.000 | 1.000 | −1.807 | 0.221 | −0.288 | 0.035 | 0.000 |
| closed_list | 1.000 | −6.046 | 0.814 | −0.438 | 0.059 | 0.000 | 1.000 | −6.092 | 0.827 | −0.441 | 0.060 | 0.000 |
| advanced | 1.000 | 5.356 | 0.828 | 0.285 | 0.044 | 1.000 | 1.000 | 5.219 | 0.828 | 0.277 | 0.044 | 1.000 |
| resour_rich | 1.000 | 9.990 | 1.401 | 0.284 | 0.040 | 1.000 | 1.000 | 10.002 | 1.403 | 0.284 | 0.040 | 1.000 |
| religion_frac | 1.000 | −10.079 | 1.533 | −0.322 | 0.049 | 0.000 | 1.000 | −10.405 | 1.448 | −0.332 | 0.046 | 0.000 |
| German_LE | 1.000 | 14.844 | 1.634 | 0.584 | 0.064 | 1.000 | 1.000 | 14.599 | 1.569 | 0.574 | 0.062 | 1.000 |
| ethnic_frac | 1.000 | −18.058 | 2.573 | −0.468 | 0.067 | 0.000 | 1.000 | −19.108 | 1.915 | −0.495 | 0.050 | 0.000 |
| polcon5 | 1.000 | −15.752 | 3.224 | −0.169 | 0.035 | 0.000 | 1.000 | −16.092 | 3.181 | −0.173 | 0.034 | 0.000 |
| fed | 1.000 | −5.309 | 1.187 | −0.308 | 0.069 | 0.000 | 0.997 | −4.919 | 1.250 | −0.285 | 0.072 | 0.000 |
| gov_frac | 0.995 | 9.971 | 1.914 | 0.368 | 0.071 | 1.000 | 0.991 | 10.114 | 1.668 | 0.374 | 0.062 | 1.000 |
| pub_bal | 0.933 | −0.582 | 0.176 | −0.417 | 0.126 | 0.000 | 0.907 | −0.579 | 0.195 | −0.415 | 0.140 | 0.000 |
| vot_turn | 0.823 | 0.058 | 0.035 | 0.103 | 0.062 | 1.000 | 0.672 | 0.044 | 0.036 | 0.078 | 0.063 | 1.000 |
| dr_nat | 0.779 | −1.33 | 0.863 | −0.081 | 0.053 | 0.000 | 0.791 | −1.439 | 0.889 | −0.088 | 0.054 | 0.000 |

**Table A2.** *Cont.*

| Model Prior | Beta-Binomial | | | | | | Dilution | | | | | |
|---|---|---|---|---|---|---|---|---|---|---|---|---|
| g Prior | Risk Inflation Criterion | | | | | | Unit Information Prior | | | | | |
| Variable | PIP | PM | PSD | SPM | SPSD | P(+) | PIP | PM | PSD | SPM | SPSD | P(+) |
| debt_pub | 0.555 | 0.017 | 0.017 | 0.068 | 0.071 | 1.000 | 0.427 | 0.013 | 0.017 | 0.054 | 0.070 | 1.000 |
| language_frac | 0.220 | −1.379 | 3.047 | −0.039 | 0.087 | 0.014 | 0.032 | −0.207 | 1.289 | −0.006 | 0.037 | 0.010 |
| polcon3 | 0.204 | 0.774 | 1.769 | 0.013 | 0.031 | 1.000 | 0.126 | 0.473 | 1.431 | 0.008 | 0.025 | 1.000 |
| no_part | 0.129 | 0.082 | 0.262 | 0.015 | 0.047 | 1.000 | 0.038 | 0.035 | 0.208 | 0.006 | 0.037 | 1.000 |
| bud_bal | 0.125 | −0.056 | 0.178 | −0.038 | 0.122 | 0.000 | 0.096 | −0.064 | 0.198 | −0.044 | 0.135 | 0.000 |
| gdpgr | 0.087 | −0.006 | 0.028 | −0.003 | 0.014 | 0.000 | 0.085 | −0.007 | 0.028 | −0.003 | 0.014 | 0.000 |
| yrs_elec | 0.082 | −0.014 | 0.059 | −0.002 | 0.010 | 0.000 | 0.096 | −0.015 | 0.062 | −0.003 | 0.011 | 0.000 |
| unemployment | 0.082 | −0.007 | 0.034 | −0.004 | 0.018 | 0.010 | 0.042 | −0.004 | 0.025 | −0.002 | 0.013 | 0.007 |
| dist_house | 0.077 | −0.001 | 0.007 | −0.004 | 0.022 | 0.121 | 0.023 | −0.000 | 0.003 | −0.001 | 0.011 | 0.029 |
| rr_nat | 0.062 | 0.050 | 0.284 | 0.003 | 0.014 | 1.000 | 0.026 | 0.016 | 0.167 | 0.001 | 0.008 | 0.995 |
| dr | 0.060 | 0.040 | 0.244 | 0.002 | 0.013 | 0.984 | 0.035 | 0.026 | 0.191 | 0.001 | 0.010 | 0.993 |
| x2009 | 0.050 | −0.025 | 0.200 | −0.001 | 0.008 | 0.019 | 0.061 | −0.027 | 0.209 | −0.001 | 0.008 | 0.023 |
| e_union | 0.045 | −0.015 | 0.224 | −0.001 | 0.012 | 0.292 | 0.013 | 0.000 | 0.104 | 0.000 | 0.006 | 0.430 |
| elec_year | 0.043 | 0.006 | 0.08 | 0.000 | 0.005 | 0.913 | 0.056 | 0.008 | 0.091 | 0.000 | 0.006 | 0.934 |
| x2010 | 0.042 | −0.007 | 0.125 | −0.000 | 0.005 | 0.010 | 0.051 | −0.009 | 0.140 | −0.000 | 0.006 | 0.017 |
| inflation | 0.040 | 0.001 | 0.022 | 0.000 | 0.006 | 0.807 | 0.040 | 0.001 | 0.022 | 0.000 | 0.006 | 0.936 |
| bbr | 0.039 | 0.010 | 0.167 | 0.000 | 0.007 | 0.914 | 0.028 | 0.008 | 0.139 | 0.000 | 0.006 | 0.968 |
| Burn-ins | 100,000 | | | | | | | | | | | |
| Iterations | 1 m | | | | | | | | | | | |
| Cor PMP | 0.9995 | | | | | | 0.9996 | | | | | |

## Appendix C. Jointness Measures

**Table A3.** Jointness measures under UIP and uniform priors-LS (above diagonal) and DW (below diagonal) measures.

| JDW \ JLS | dr_nat | er_nat | rr_nat | bbr_nat | English_LE | French_LE | Socialist_LE | no_part | polcon3 | polcon5 | gdpgr | unemployment | dist_house | elec_sys | e_union | fed | cur_union | bud_bal | pub_bal | debt_pub | reg_age | checks_bal | inflation | yrs_elec | closed_list | advanced | resour_rich | vot_turn | gov_frac | religion_frac | ethnic_frac | language_frac | Germana_LE | elec_year | bbr | DR | x2009 | x2010 |
|---|---|---|---|---|---|---|---|---|---|---|---|---|---|---|---|---|---|---|---|---|---|---|---|---|---|---|---|---|---|---|---|---|---|---|---|---|---|---|
| dr_nat | x | 1,1 | -3,3 | 1,1 | 1,1 | 1,1 | 1,1 | -2,2 | -1,9 | 1,1 | -2,9 | -3,4 | -3,1 | 1,1 | -3,4 | 1,1 | 1,1 | -2,2 | 1,0 | -1,0 | 1,1 | 1,1 | -3,6 | -2,9 | 1,1 | 1,1 | 1,1 | 0,6 | 1,1 | 1,1 | 1,1 | -1,5 | 1,1 | -3,6 | -3,6 | -3,0 | -3,3 | -3,6 |
| er_nat | -0,8 | x | -3,1 | 8,3 | 8,3 | 8,3 | 8,3 | -2,2 | -1,8 | 7,7 | -2,9 | -2,8 | -3,0 | 8,3 | -3,4 | 7,2 | 8,3 | -2,0 | 2,5 | 0,0 | 8,3 | 8,3 | -3,5 | -2,8 | 8,3 | 8,3 | 8,3 | 1,2 | 5,6 | 8,2 | 7,3 | -1,6 | 8,3 | -3,6 | -3,5 | -3,1 | -3,4 | -3,6 |
| rr_nat | -0,6 | +inf | x | -3,1 | -3,1 | -3,1 | -3,1 | -3,3 | -3,3 | -3,1 | -3,6 | -3,1 | -3,6 | -3,1 | -4,0 | -3,1 | -3,1 | -3,4 | -3,1 | -3,0 | -3,1 | -3,1 | -4,3 | -3,6 | -3,1 | -3,1 | -3,1 | -3,2 | -3,1 | -3,1 | -3,1 | -3,0 | -3,1 | -4,2 | -3,8 | -3,8 | -3,6 | -4,0 |
| bbr_nat | nan | nan | nan | x | 0,0 | 0,0 | 0,0 | -2,2 | -1,8 | 8,5 | -2,9 | -2,8 | -3,0 | 0,0 | -3,4 | 7,6 | 0,0 | -2,0 | 2,5 | 0,0 | 0,0 | 0,0 | -3,5 | -2,8 | 0,0 | 0,0 | 0,0 | 1,2 | 5,7 | 10,2 | 7,8 | -1,6 | 0,0 | -3,6 | -3,5 | -3,1 | -3,4 | -3,6 |
| English_LE | nan | nan | nan | nan | x | 0,0 | 0,0 | -2,2 | -1,8 | 8,5 | -2,9 | -2,8 | -3,0 | 0,0 | -3,4 | 7,6 | 0,0 | -2,0 | 2,5 | 0,0 | 0,0 | 0,0 | -3,5 | -2,8 | 0,0 | 0,0 | 0,0 | 1,2 | 5,7 | 10,2 | 7,8 | -1,6 | 0,0 | -3,6 | -3,5 | -3,1 | -3,4 | -3,6 |
| French_LE | nan | nan | nan | nan | nan | x | 0,0 | -2,2 | -1,8 | 8,5 | -2,9 | -2,8 | -3,0 | 0,0 | -3,4 | 7,6 | 0,0 | -2,0 | 2,5 | 0,0 | 0,0 | 0,0 | -3,5 | -2,8 | 0,0 | 0,0 | 0,0 | 1,2 | 5,7 | 10,2 | 7,8 | -1,6 | 0,0 | -3,6 | -3,5 | -3,1 | -3,4 | -3,6 |
| Socialist_LE | nan | nan | nan | nan | nan | nan | x | -2,2 | -1,8 | 8,5 | -2,9 | -2,8 | -3,0 | 0,0 | -3,4 | 7,6 | 0,0 | -2,0 | 2,5 | 0,0 | 0,0 | 0,0 | -3,5 | -2,8 | 0,0 | 0,0 | 0,0 | 1,2 | 5,7 | 10,2 | 7,8 | -1,6 | 0,0 | -3,6 | -3,5 | -3,1 | -3,4 | -3,6 |
| no_part | 0,1 | 2,1 | 0,0 | nan | nan | nan | nan | x | -3,0 | -2,2 | -3,4 | -3,5 | -3,2 | -2,2 | -3,6 | -2,2 | -2,2 | -2,9 | -2,2 | -2,6 | -2,2 | -2,2 | -3,7 | -3,3 | -2,2 | -2,2 | -2,2 | -2,4 | -2,2 | -2,2 | -2,2 | -2,9 | -2,2 | -3,6 | -4,2 | -3,6 | -3,9 | -3,8 |
| polcon3 | 0,0 | -1,3 | 0,2 | nan | nan | nan | nan | -0,3 | x | -1,8 | -3,0 | -3,4 | -3,4 | -1,8 | -3,5 | -1,8 | -1,8 | -2,9 | -1,8 | -1,8 | -1,8 | -1,8 | -3,6 | -3,3 | -1,8 | -1,8 | -1,8 | -1,8 | -1,8 | -1,8 | -1,8 | -2,5 | -1,8 | -4,0 | -3,7 | -3,3 | -3,6 | -3,7 |
| polcon5 | -0,3 | nan | +inf | nan | nan | nan | nan | 0,0 | +inf | x | -2,9 | -2,8 | -3,0 | 8,5 | -3,4 | 7,2 | 8,5 | -2,0 | 2,5 | 0,0 | 8,5 | 8,5 | -3,5 | -2,8 | 8,5 | 8,5 | 8,5 | 1,2 | 5,6 | 8,3 | 7,4 | -1,6 | 8,5 | -3,6 | -3,5 | -3,1 | -3,4 | -3,6 |
| gdpgr | 0,0 | -1,7 | 0,1 | nan | nan | nan | nan | 0,0 | 0,3 | +inf | x | -3,5 | -3,7 | -2,9 | -4,1 | -2,9 | -2,9 | -2,7 | -3,0 | -2,9 | -2,9 | -2,9 | -3,9 | -3,3 | -2,9 | -2,9 | -2,9 | -3,0 | -2,9 | -2,9 | -2,9 | -3,2 | -2,9 | -3,8 | -4,0 | -3,6 | -2,7 | -4,1 |
| unemployment | -1,2 | -0,6 | 0,5 | nan | nan | nan | nan | -0,3 | -0,3 | -0,8 | 0,1 | x | -3,0 | -2,8 | -3,8 | -2,8 | -2,8 | -3,4 | -2,8 | -2,4 | -2,8 | -2,8 | -4,1 | -3,6 | -2,8 | -2,8 | -2,8 | -3,0 | -2,8 | -2,8 | -2,8 | -3,7 | -2,8 | -4,1 | -3,8 | -3,7 | -3,9 | -3,9 |
| dist_house | -0,3 | -0,9 | 0,3 | nan | nan | nan | nan | 0,4 | 0,1 | +inf | 0,0 | 0,5 | x | -3,0 | -3,6 | -3,0 | -3,0 | -3,2 | -3,0 | -2,7 | -3,0 | -3,0 | -4,1 | -3,5 | -3,0 | -3,0 | -3,0 | -3,1 | -3,0 | -3,0 | -3,0 | -3,5 | -3,0 | -4,1 | -3,8 | -4,2 | -4,2 | -4,0 |
| elec_sys | nan | nan | nan | nan | nan | nan | nan | nan | nan | nan | nan | nan | nan | x | -3,4 | 7,6 | 0,0 | -2,0 | 2,5 | 0,0 | 0,0 | 0,0 | -3,5 | -2,8 | 0,0 | 0,0 | 0,0 | 1,2 | 5,7 | 10,2 | 7,8 | -1,6 | 0,0 | -3,6 | -3,5 | -3,1 | -3,4 | -3,6 |
| e_union | 0,1 | 0,1 | 0,0 | nan | nan | nan | nan | 0,0 | 0,0 | +inf | -0,3 | -0,1 | 0,3 | nan | x | -3,4 | -3,4 | -3,6 | -3,4 | -3,6 | -3,4 | -3,4 | -4,2 | -4,0 | -3,4 | -3,4 | -3,4 | -3,4 | -3,4 | -3,4 | -3,4 | -3,3 | -3,4 | -4,1 | -4,1 | -3,8 | -4,6 | -3,9 |
| fed | 0,6 | -inf | -0,3 | nan | nan | nan | nan | +inf | -0,2 | -inf | 0,1 | +inf | -0,4 | nan | -1,2 | x | 7,6 | -2,0 | 2,5 | 0,0 | 7,6 | 7,6 | -3,5 | -2,8 | 7,6 | 7,6 | 7,6 | 1,2 | 5,5 | 7,5 | 7,0 | -1,6 | 7,6 | -3,6 | -3,5 | -3,1 | -3,4 | -3,6 |
| cur_union | nan | nan | nan | nan | nan | nan | nan | nan | nan | nan | nan | nan | nan | nan | nan | nan | x | -2,0 | 2,5 | 0,0 | 0,0 | 0,0 | -3,5 | -2,8 | 0,0 | 0,0 | 0,0 | 1,2 | 5,7 | 10,2 | 7,8 | -1,6 | 0,0 | -3,6 | -3,5 | -3,1 | -3,4 | -3,6 |
| bud_bal | -0,4 | 1,3 | -0,2 | nan | nan | nan | nan | 0,1 | -0,3 | +inf | 0,4 | -0,3 | 0,1 | nan | 0,3 | -2,8 | nan | x | -3,1 | -1,9 | -2,0 | -2,0 | -3,8 | -3,2 | -2,0 | -2,0 | -2,0 | -2,3 | -2,0 | -2,0 | -2,0 | -2,8 | -2,0 | -3,9 | -3,9 | -3,6 | -3,5 | -3,8 |
| pub_bal | 0,6 | -0,8 | 0,3 | nan | nan | nan | nan | -0,2 | 0,3 | nan | -0,5 | 0,5 | -0,3 | nan | -0,3 | 3,2 | nan | nan | x | -0,1 | 2,5 | 2,5 | -3,5 | -2,8 | 2,5 | 2,5 | 2,5 | 1,1 | 2,5 | 2,5 | 2,5 | -1,6 | 2,5 | -3,6 | -3,5 | -3,1 | -3,4 | -3,6 |
| debt_pub | -3,9 | 0,9 | 0,5 | nan | nan | nan | nan | -0,6 | 0,3 | 1,5 | -0,1 | 1,0 | 0,8 | nan | -0,1 | 0,2 | nan | 0,4 | -0,7 | x | 0,0 | 0,0 | -3,5 | -2,8 | 0,0 | 0,0 | 0,0 | -0,2 | 0,0 | 0,0 | 0,0 | -2,7 | 0,0 | -3,7 | -3,4 | -3,2 | -3,5 | -3,6 |
| reg_age | nan | nan | nan | nan | nan | nan | nan | nan | nan | nan | nan | nan | nan | nan | nan | nan | nan | nan | nan | nan | x | 0,0 | -3,5 | -2,8 | 0,0 | 0,0 | 0,0 | 1,2 | 5,7 | 10,2 | 7,8 | -1,6 | 0,0 | -3,6 | -3,5 | -3,1 | -3,4 | -3,6 |
| checks_bal | nan | nan | nan | nan | nan | nan | nan | nan | nan | nan | nan | nan | nan | nan | nan | nan | nan | nan | nan | nan | nan | x | -3,5 | -2,8 | 0,0 | 0,0 | 0,0 | 1,2 | 5,7 | 10,2 | 7,8 | -1,6 | 0,0 | -3,6 | -3,5 | -3,1 | -3,4 | -3,6 |
| inflation | -0,1 | +inf | 0,3 | nan | nan | nan | nan | 0,0 | 0,0 | +inf | 0,5 | 0,0 | 0,1 | nan | 0,1 | +inf | nan | 0,2 | -0,2 | 0,1 | nan | nan | x | -3,6 | -3,5 | -3,5 | -3,5 | -3,5 | -3,5 | -3,5 | -3,5 | -3,7 | -3,5 | -4,1 | -4,4 | -4,1 | -4,0 | -4,1 |
| yrs_elec | -0,3 | +inf | 0,2 | nan | nan | nan | nan | -0,1 | -0,4 | +inf | 0,2 | 0,5 | 0,2 | nan | 0,2 | +inf | nan | -0,1 | 0,0 | 0,3 | nan | nan | -0,1 | x | -2,8 | -2,8 | -2,8 | -2,9 | -2,8 | -2,8 | -2,8 | -3,1 | -2,8 | -4,2 | -4,1 | -3,8 | -3,5 | -3,9 |
| closed_list | nan | nan | nan | nan | nan | nan | nan | nan | nan | nan | nan | nan | nan | nan | nan | nan | nan | nan | nan | nan | nan | nan | nan | nan | x | 0,0 | 0,0 | 1,2 | 5,7 | 10,2 | 7,8 | -1,6 | 0,0 | -3,6 | -3,5 | -3,1 | -3,4 | -3,6 |
| advanced | nan | nan | nan | nan | nan | nan | nan | nan | nan | nan | nan | nan | nan | nan | nan | nan | nan | nan | nan | nan | nan | nan | nan | nan | nan | x | 0,0 | 1,2 | 5,7 | 10,2 | 7,8 | -1,6 | 0,0 | -3,6 | -3,5 | -3,1 | -3,4 | -3,6 |
| resour_rich | nan | nan | nan | nan | nan | nan | nan | nan | nan | nan | nan | nan | nan | nan | nan | nan | nan | nan | nan | nan | nan | nan | nan | nan | nan | nan | x | 1,2 | 5,7 | 10,2 | 7,8 | -1,6 | 0,0 | -3,6 | -3,5 | -3,1 | -3,4 | -3,6 |
| vot_turn | 0,5 | -0,7 | -0,2 | nan | nan | nan | nan | -0,5 | 0,2 | -inf | -0,2 | -0,6 | -0,3 | nan | 0,1 | +inf | nan | -0,6 | 0,8 | 0,0 | nan | nan | 0,0 | 0,0 | nan | nan | nan | x | 1,2 | 1,2 | 1,2 | -1,3 | 1,2 | -3,6 | -3,5 | -3,3 | -3,4 | -3,6 |
| gov_frac | 1,3 | -inf | -0,7 | nan | nan | nan | nan | -33,3 | -1,3 | -inf | -0,3 | 0,2 | 0,0 | nan | -0,5 | -inf | nan | -1,4 | 1,8 | 0,2 | nan | nan | -0,2 | 0,7 | nan | nan | nan | 1,1 | x | 5,7 | 5,6 | -1,6 | 5,7 | -3,6 | -3,5 | -3,1 | -3,4 | -3,6 |
| religion_frac | nan | nan | nan | nan | nan | nan | nan | nan | nan | nan | nan | nan | nan | nan | nan | nan | nan | nan | nan | nan | nan | nan | nan | nan | nan | nan | nan | nan | nan | x | 5,7 | -1,6 | 10,2 | -3,6 | -3,5 | -3,1 | -3,4 | -3,6 |
| ethnic_frac | 2,1 | -inf | -30,3 | nan | nan | nan | nan | +inf | 0,7 | -inf | +inf | -7,0 | -3,4 | nan | +inf | -inf | nan | +inf | -inf | +inf | nan | nan | +inf | +inf | nan | nan | nan | -26,0 | -21,6 | nan | x | -1,6 | 7,8 | -3,6 | -3,5 | -3,1 | -3,4 | -3,6 |
| language_frac | 0,8 | -3,5 | 0,3 | nan | nan | nan | nan | -0,4 | -0,1 | -2,1 | -0,1 | -0,6 | -0,4 | nan | 0,4 | 1,6 | nan | -0,5 | 0,8 | -1,5 | nan | nan | 0,0 | 0,0 | nan | nan | nan | 2,2 | 1,7 | nan | -inf | x | -1,6 | -3,6 | -3,6 | -3,6 | -3,5 | -3,6 |
| Germana_LE | nan | nan | nan | nan | nan | nan | nan | nan | nan | nan | nan | nan | nan | nan | nan | nan | nan | nan | nan | nan | nan | nan | nan | nan | nan | nan | nan | nan | nan | nan | nan | nan | x | -3,6 | -3,5 | -3,1 | -3,4 | -3,6 |
| elec_year | 0,0 | +inf | 0,1 | nan | nan | nan | nan | 0,3 | -0,1 | +inf | -0,1 | -0,1 | -0,1 | nan | -0,1 | 1,9 | nan | 0,1 | -0,2 | -0,1 | nan | nan | 0,3 | -0,1 | nan | nan | nan | 0,0 | -0,3 | nan | +inf | 0,1 | nan | x | -4,7 | -3,9 | -4,5 | -4,2 |
| bbr | -0,1 | +inf | 0,2 | nan | nan | nan | nan | -0,2 | 0,1 | -2,7 | -0,2 | -0,1 | -0,3 | nan | 0,2 | +inf | nan | -0,2 | 0,3 | 0,0 | nan | nan | -0,3 | -0,2 | nan | nan | nan | 0,1 | -0,3 | nan | +inf | 0,1 | nan | 0,1 | x | -4,0 | -4,3 | -4,1 |
| DR | 0,4 | +inf | -0,1 | nan | nan | nan | nan | -0,1 | 0,0 | +inf | 0,0 | 0,1 | 0,0 | nan | 0,2 | 1,9 | nan | -0,2 | 0,1 | 0,1 | nan | nan | -0,2 | 0,0 | nan | nan | nan | -0,4 | 0,4 | nan | +inf | -0,3 | nan | 0,1 | 0,1 | x | -4,0 | -4,2 |
| x2009 | 0,2 | +inf | 0,0 | nan | nan | nan | nan | 0,0 | 0,2 | +inf | 1,2 | -0,2 | -0,3 | nan | 0,1 | -1,0 | nan | 0,2 | -0,3 | -0,2 | nan | nan | -0,1 | 0,0 | nan | nan | nan | 0,0 | 0,0 | nan | +inf | 0,1 | nan | 0,0 | -0,2 | 0,3 | x | -4,0 |
| x2010 | 0,2 | 0,0 | -0,1 | nan | nan | nan | nan | 0,1 | -0,1 | +inf | -0,3 | -0,1 | 0,0 | nan | -0,2 | +inf | nan | -0,2 | 0,3 | -0,1 | nan | nan | -0,3 | 0,2 | nan | nan | nan | 0,1 | 1,8 | nan | +inf | 0,0 | nan | 0,5 | -0,5 | -0,1 | 0,0 | x |

## Appendix D. Overall and Country Specific Descriptive Statistics

**Table A4.** Descriptive statistics.

| *Variable* | **N** | **Min** | **Max** | **Avg** | **Std. Dev.** |
|---|---|---|---|---|---|
| *cen_gov_exp* | 300 | 13.25 | 63.80 | 29.20 | 6.91 |
| *pub_bal* | 300 | −32.18 | 18.46 | −1.91 | 4.95 |
| *bud_bal* | 300 | −29.20 | 19.66 | −1.43 | 4.74 |
| *debt_pub* | 300 | 3.66 | 125.76 | 51.15 | 28.20 |
| *gdpgr* | 300 | −17.70 | 10.99 | 2.06 | 3.38 |
| *unemployment* | 300 | 2.20 | 25.00 | 7.86 | 3.68 |
| *inflation* | 300 | −2.75 | 14.03 | 2.75 | 1.87 |
| *reg_age* | 300 | 14.50 | 207.50 | 68.14 | 48.65 |
| *ethnic_frac* | 300 | 0.05 | 0.59 | 0.24 | 0.18 |
| *religion_frac* | 300 | 0.09 | 0.82 | 0.42 | 0.22 |
| *language_frac* | 300 | 0.02 | 0.64 | 0.26 | 0.20 |
| *gov_frac* | 300 | 0.00 | 0.83 | 0.37 | 0.26 |
| *polcon3* | 300 | 0.12 | 0.72 | 0.47 | 0.12 |
| *polcon5* | 300 | 0.34 | 0.89 | 0.77 | 0.07 |
| *vot_turn* | 300 | 40.57 | 94.85 | 71.01 | 12.26 |
| *no_part* | 300 | 1.00 | 6.00 | 2.53 | 1.24 |
| *checks_bal* | 300 | 2.00 | 8.00 | 4.27 | 1.10 |
| *dist_house* | 300 | 0.90 | 120.00 | 13.89 | 22.52 |

Source: own compilation.

**Table A5.** Summary of observations-individual countries.

| **Country avg.** | Australia | Austria | Belgium | Bulgaria | Czech Republic | Denmark | Estonia | Finland | France | Germany | Hungary | Ireland | Israel |
|---|---|---|---|---|---|---|---|---|---|---|---|---|---|
| *cen_gov_exp* | 25.25 | 34.48 | 28.58 | 26.83 | 30.50 | 38.07 | 27.94 | 25.96 | 23.46 | 14.08 | 33.03 | 37.15 | 39.40 |
| *pub_bal* | −0.95 | −2.57 | −1.90 | 0.39 | −3.79 | 0.83 | 0.50 | 1.75 | −3.92 | −2.25 | −5.78 | −5.53 | −4.82 |
| *bud_bal* | −0.38 | −2.07 | −1.62 | 0.63 | −3.63 | 0.98 | 0.86 | 2.27 | −3.81 | −1.47 | −4.66 | −5.15 | −4.67 |
| *debt_pub* | 15.75 | 71.45 | 98.30 | 29.20 | 31.20 | 41.07 | 5.61 | 41.78 | 70.43 | 68.27 | 67.10 | 51.35 | 79.30 |
| *gdpgr* | 3.06 | 1.57 | 1.31 | 3.66 | 2.93 | 0.58 | 4.27 | 1.65 | 1.11 | 1.16 | 1.66 | 2.30 | 3.59 |

**Table A5.** *Cont.*

| Country avg. | Australia | Austria | Belgium | Bulgaria | Czech Republic | Denmark | Estonia | Finland | France | Germany | Hungary | Ireland | Israel |
|---|---|---|---|---|---|---|---|---|---|---|---|---|---|
| *unemployment* | 5.33 | 4.41 | 7.75 | 11.14 | 7.00 | 5.38 | 10.11 | 8.13 | 9.04 | 8.39 | 8.08 | 7.71 | 10.07 |
| *inflation* | 2.83 | 2.08 | 2.21 | 5.12 | 2.47 | 2.09 | 4.19 | 1.97 | 1.91 | 1.70 | 5.24 | 2.33 | 2.26 |
| *reg_age* | 106.50 | 61.50 | 113.50 | 17.50 | 14.50 | 106.50 | 16.50 | 90.50 | 61.50 | 17.50 | 17.50 | 85.50 | 59.50 |
| *ethnic_frac* | 0.09 | 0.11 | 0.56 | 0.40 | 0.32 | 0.08 | 0.51 | 0.13 | 0.10 | 0.17 | 0.15 | 0.12 | 0.34 |
| *religion_frac* | 0.82 | 0.41 | 0.21 | 0.60 | 0.66 | 0.23 | 0.50 | 0.25 | 0.40 | 0.66 | 0.52 | 0.15 | 0.35 |
| *language_frac* | 0.33 | 0.15 | 0.54 | 0.30 | 0.32 | 0.10 | 0.49 | 0.14 | 0.12 | 0.16 | 0.03 | 0.03 | 0.55 |
| *gov_frac* | 0.16 | 0.44 | 0.80 | 0.41 | 0.36 | 0.48 | 0.61 | 0.64 | 0.18 | 0.40 | 0.14 | 0.18 | 0.70 |
| *polcon3* | 0.44 | 0.49 | 0.71 | 0.48 | 0.46 | 0.31 | 0.53 | 0.54 | 0.51 | 0.46 | 0.37 | 0.46 | 0.58 |
| *polcon5* | 0.86 | 0.75 | 0.89 | 0.61 | 0.74 | 0.73 | 0.77 | 0.77 | 0.87 | 0.85 | 0.74 | 0.76 | 0.78 |
| *vot_turn* | 94.29 | 80.71 | 90.71 | 61.01 | 62.62 | 86.63 | 60.21 | 66.02 | 60.41 | 76.10 | 66.88 | 65.56 | 65.05 |
| *dist_house* | 0.90 | 20.30 | 13.63 | 7.72 | 16.08 | 10.50 | 9.20 | 13.33 | 1.00 | 1.90 | 8.88 | 4.00 | 120.00 |
| *no_part* | 1.58 | 2.00 | 5.42 | 2.75 | 2.33 | 2.67 | 2.83 | 3.92 | 2.17 | 2.00 | 1.67 | 2.33 | 5.00 |
| *checks_bal* | 4.58 | 4.00 | 4.08 | 2.67 | 5.50 | 5.25 | 3.25 | 4.25 | 4.17 | 4.58 | 3.50 | 5.42 | 4.33 |
| *French_LO* | 0 | 0 | 1 | 0 | 0 | 0 | 0 | 0 | 1 | 0 | 0 | 0 | 0 |
| *Socialist_LO* | 0 | 0 | 0 | 1 | 1 | 0 | 1 | 0 | 0 | 0 | 1 | 0 | 0 |
| *German_LO* | 0 | 1 | 0 | 0 | 0 | 0 | 0 | 0 | 0 | 1 | 0 | 0 | 0 |
| *English_LO* | 1 | 0 | 0 | 0 | 0 | 0 | 0 | 0 | 0 | 0 | 0 | 1 | 1 |
| *cur_union* | 0 | 1 | 1 | 0 | 0 | 0 | 1 | 1 | 1 | 1 | 0 | 1 | 0 |
| *e_union* | 0 | 1 | 1 | 1 | 1 | 1 | 1 | 1 | 1 | 1 | 1 | 1 | 0 |
| *fed* | 1 | 1 | 1 | 0 | 0 | 0 | 0 | 0 | 0 | 1 | 0 | 0 | 0 |
| *advanced* | 1 | 1 | 1 | 0 | 1 | 1 | 1 | 1 | 1 | 1 | 0 | 1 | 1 |
| *resour_rich* | 0 | 0 | 0 | 0 | 0 | 0 | 0 | 0 | 0 | 0 | 0 | 0 | 0 |
| *elec_sys* | 1 | 3 | 3 | 3 | 3 | 3 | 3 | 3 | 1 | 2 | 2 | 3 | 3 |
| *closed_list* | 0 | 1 | 1 | 1 | 0 | 0 | 0 | 0 | 0 | 1 | 0 | 0 | 1 |
| *BBR* | 1 | 1 | 1 | 1 | 1 | 1 | 1 | 1 | 1 | 1 | 1 | 1 | 1 |
| *DR* | 1 | 1 | 1 | 1 | 1 | 1 | 1 | 1 | 1 | 1 | 1 | 1 | 0 |
| *BBR_nat* | 1 | 1 | 0 | 1 | 0 | 1 | 1 | 1 | 0 | 1 | 1 | 0 | 1 |
| *DR_nat* | 1 | 0 | 0 | 1 | 0 | 0 | 0 | 1 | 0 | 0 | 0 | 0 | 0 |
| *RR_nat* | 1 | 0 | 0 | 0 | 0 | 1 | 0 | 0 | 1 | 0 | 0 | 0 | 0 |
| *ER_nat (2012)* | 1 | 0 | 0 | 1 | 0 | 1 | 0 | 1 | 1 | 1 | 0 | 0 | 1 |

Source: own compilation.

**Table A6.** Summary of observations-individual countries (continued).

| Country avg. | Italy | Latvia | Luxembourg | Netherlands | Norway | Poland | Portugal | Slovenia | Spain | Sweden | United Kingdom | United States | All |
|---|---|---|---|---|---|---|---|---|---|---|---|---|---|
| cen_gov_exp | 27.63 | 21.49 | 29.65 | 26.73 | 35.53 | 26.23 | 35.04 | 29.38 | 19.58 | 31.01 | 40.57 | 22.51 | 29.20 |
| pub_bal | −3.45 | −2.29 | 1.34 | −2.14 | 13.06 | −4.92 | −5.64 | −2.20 | −3.37 | 0.59 | −4.93 | −5.81 | −1.91 |
| bud_bal | −2.93 | −1.39 | 0.97 | −1.63 | 13.69 | −4.30 | −4.61 | −2.37 | −1.73 | 0.70 | −4.38 | −5.09 | −1.43 |
| debt_pub | 106.73 | 20.39 | 11.32 | 54.64 | 40.02 | 47.60 | 74.47 | 31.60 | 51.73 | 42.85 | 53.25 | 73.34 | 51.15 |
| gdpgr | 0.16 | 4.23 | 2.33 | 1.11 | 1.59 | 3.80 | 0.21 | 2.15 | 1.59 | 2.16 | 1.51 | 1.76 | 2.06 |
| unemployment | 8.07 | 11.86 | 4.27 | 4.09 | 3.57 | 13.55 | 8.46 | 6.47 | 13.87 | 7.11 | 6.09 | 6.51 | 7.86 |
| inflation | 2.41 | 5.03 | 2.68 | 2.22 | 1.82 | 2.79 | 2.44 | 3.60 | 2.84 | 1.77 | 2.41 | 2.37 | 2.75 |
| reg_age | 61.50 | 14.50 | 117.50 | 110.50 | 107.50 | 18.50 | 31.50 | 16.50 | 30.50 | 96.50 | 122.50 | 207.50 | 68.14 |
| ethnic_frac | 0.11 | 0.59 | 0.53 | 0.11 | 0.06 | 0.12 | 0.05 | 0.22 | 0.42 | 0.06 | 0.12 | 0.49 | 0.24 |
| religion_frac | 0.30 | 0.56 | 0.09 | 0.72 | 0.20 | 0.17 | 0.14 | 0.29 | 0.45 | 0.23 | 0.69 | 0.82 | 0.42 |
| language_frac | 0.11 | 0.58 | 0.64 | 0.51 | 0.07 | 0.05 | 0.02 | 0.22 | 0.41 | 0.20 | 0.05 | 0.25 | 0.26 |
| gov_frac | 0.11 | 0.63 | 0.48 | 0.58 | 0.48 | 0.29 | 0.08 | 0.56 | 0.01 | 0.52 | 0.04 | 0.00 | 0.37 |
| polcon3 | 0.40 | 0.51 | 0.51 | 0.56 | 0.53 | 0.46 | 0.38 | 0.40 | 0.37 | 0.45 | 0.38 | 0.40 | 0.47 |
| polcon5 | 0.68 | 0.77 | 0.77 | 0.76 | 0.77 | 0.74 | 0.74 | 0.75 | 0.85 | 0.76 | 0.74 | 0.85 | 0.77 |
| vot_turn | 81.43 | 65.35 | 90.14 | 78.26 | 76.43 | 48.27 | 61.84 | 64.51 | 72.72 | 81.97 | 61.80 | 56.44 | 71.01 |
| dist_house | 14.04 | 20.00 | 15.00 | 8.30 | 9.45 | 11.66 | 10.50 | 10.50 | 6.80 | 11.60 | 1.00 | 1.00 | 13.89 |
| no_part | 1.67 | 3.17 | 2.00 | 3.00 | 3.00 | 2.17 | 1.33 | 3.58 | 1.08 | 3.50 | 1.17 | 1.00 | 2.53 |
| checks_bal | 3.42 | 5.17 | 4.00 | 5.92 | 5.00 | 4.08 | 2.50 | 5.42 | 3.58 | 4.67 | 3.17 | 4.17 | 4.27 |
| French_LO | 1 | 0 | 1 | 1 | 0 | 0 | 1 | 0 | 1 | 0 | 0 | 0 | 0.28 |
| Socialist_LO | 0 | 1 | 0 | 0 | 0 | 1 | 0 | 1 | 0 | 0 | 0 | 0 | 0.28 |
| German_LO | 0 | 0 | 0 | 0 | 0 | 0 | 0 | 0 | 0 | 0 | 0 | 0 | 0.08 |
| English_LO | 0 | 0 | 0 | 0 | 0 | 0 | 0 | 0 | 0 | 0 | 1 | 1 | 0.20 |
| cur_union | 1 | 0 | 1 | 1 | 0 | 0 | 1 | 1 | 1 | 0 | 0 | 0 | 0.52 |
| e_union | 1 | 1 | 1 | 1 | 0 | 1 | 1 | 1 | 1 | 1 | 1 | 0 | 0.82 |
| fed | 0 | 0 | 0 | 0 | 0 | 0 | 0 | 0 | 0 | 0 | 0 | 1 | 0.20 |
| advanced | 1 | 0 | 1 | 1 | 1 | 0 | 1 | 1 | 1 | 1 | 1 | 1 | 0.84 |
| resour_rich | 0 | 0 | 0 | 0 | 1 | 0 | 0 | 0 | 0 | 0 | 0 | 0 | 0.04 |
| elec_sys | 3 | 3 | 3 | 3 | 3 | 3 | 3 | 3 | 3 | 3 | 1 | 1 | 2.57 |
| closed_list | 1 | 0 | 0 | 1 | 1 | 1 | 1 | 0 | 1 | 1 | 0 | 0 | 0.48 |
| BBR | 1 | 1 | 1 | 1 | 1 | 1 | 1 | 1 | 1 | 1 | 1 | 0 | 0.89 |
| DR | 1 | 1 | 1 | 1 | 0 | 1 | 1 | 1 | 1 | 1 | 1 | 0 | 0.83 |
| BBR_nat | 0 | 0 | 0 | 0 | 1 | 0 | 0 | 0 | 1 | 1 | 1 | 0 | 0.49 |
| DR_nat | 0 | 0 | 1 | 0 | 0 | 1 | 0 | 0 | 0 | 0 | 1 | 0 | 0.23 |
| RR_nat | 0 | 0 | 0 | 1 | 0 | 0 | 0 | 0 | 0 | 0 | 0 | 0 | 0.14 |
| ER_nat (2012) | 0 | 0 | 1 | 1 | 0 | 1 | 0 | 0 | 1 | 1 | 0 | 1 | 0.50 |

Source: own compilation.

## Appendix E. Variable Inclusion over Posterior Probability Mass

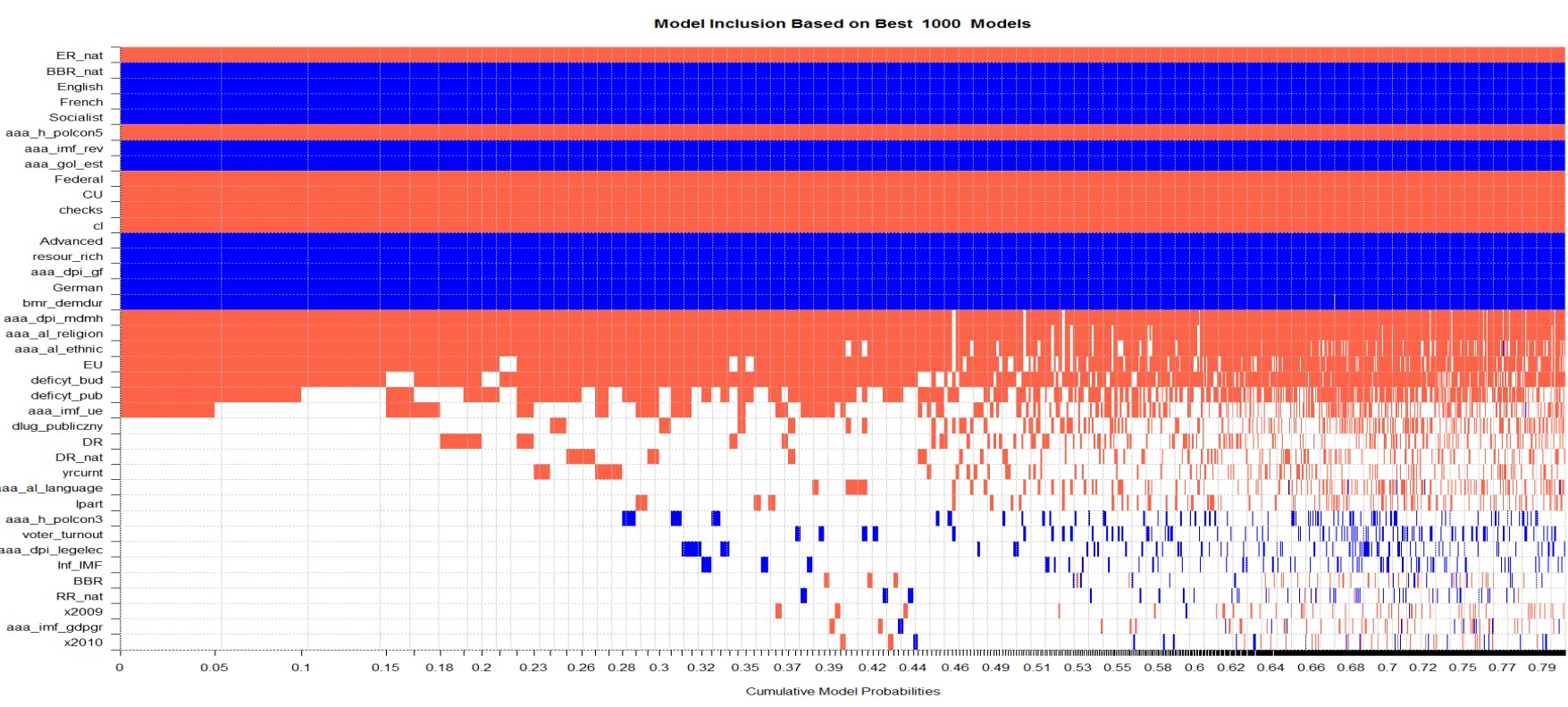

**Figure A1.** Model inclusion-1000 best models. Blue denotes variable that appears in the model with positive coefficient, red with negative coefficient, while white that the variable is excluded from the models.

## Appendix F. Variable Inclusion over Posterior Probability Mass

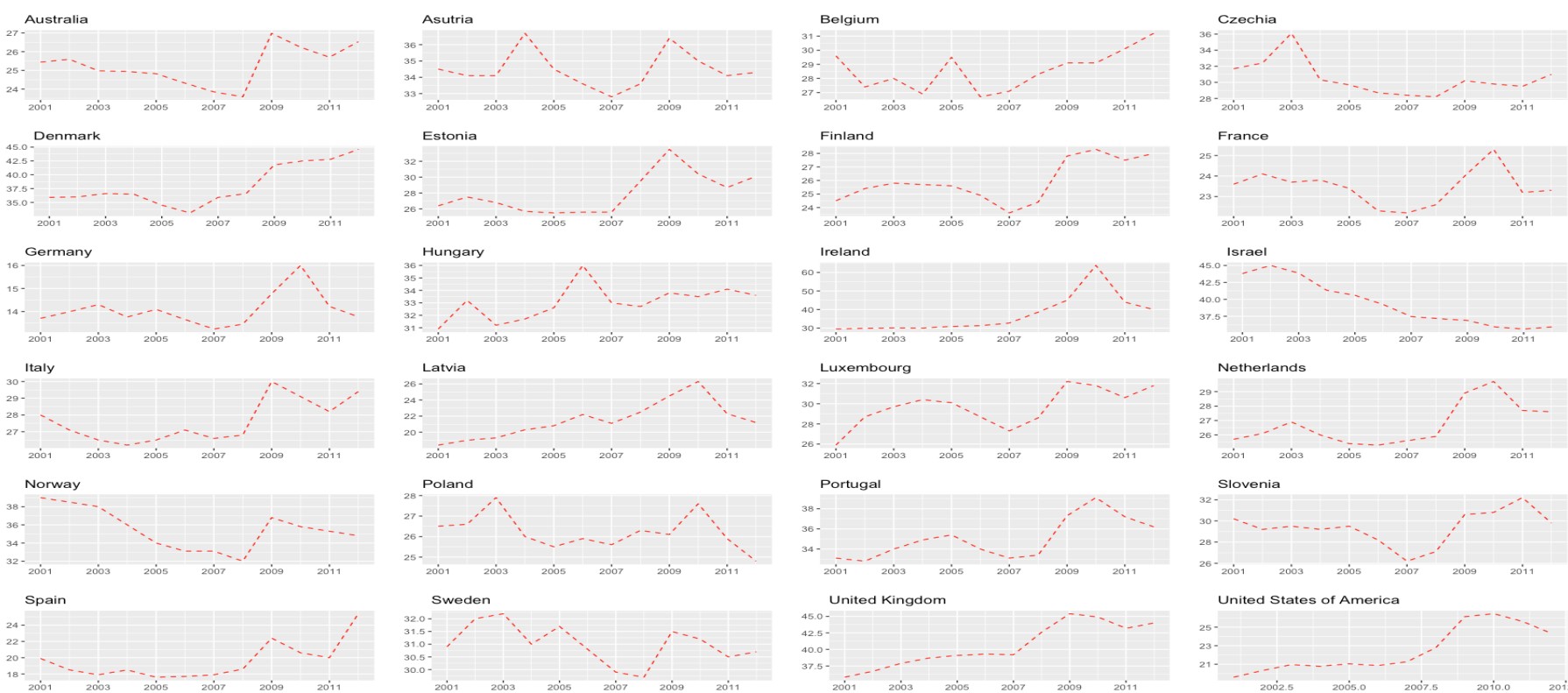

**Figure A2.** Analysis of dependent variable (Central budget expenditure)-individual countries.

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
