# Peer review of "Institutional Determinants of Budgetary Expenditures. A BMA-Based Re-Evaluation of Contemporary Theories for OECD Countries"

_sustainability, doi:10.3390/su12104104_

Round 1
Reviewer 1 Report
This paper is looking into interesting and current topic, since the management of the crisis Covid-19 of 2020 is going to influence a lot on the public expenditure of the countries.
This works has good arguments and show a complete statistical analysis.
I believe that the article needs to be thoroughly revise to allow for this:
-Do not appear an analysis of the independent variable: “government expenditures”, allow knowing its evolution between 2001-2012 and checking if there are important differences in the behaviour of this variable in the different countries, which could be explain by the independent variables.
-The paper analyses some determinants of “government expenditure”, NO all. This must be review.
-The hypothesis must be considering in section 2 “literature review and research hypotheses”. Taking into account the theoretical arguments that support them. Can not appear in the section 3 “Materials and Methods”.
-The hypothesis 3 is not enough complete in “2.1.Institutions of Power Legitimacy”. You must be complete.
-The section “3.3 BMA-Bayesian model averaging”, you have to reduce it. Because the objective of this work is not show a develop mathematics model.
-You have to explain better the selection of countries considered in the analysis. It is not clear. Why these countries, specifically? Furthermore, Bulgaria is Not a member of the OECD. Consult:
http://www.oecd.org/about/members-and-partners/
- The conclusions obtained do not correspond to those derived from other works, such as those detailed below, which you must be consider it.
Galli, E., & Rossi, S. P. (2002). Political budget cycles: the case of the Western German Länder. Public choice, 110 (3-4), 283-303.
Giuliodori, M., & Beetsma, R. (2008). On the relationship between fiscal plans in the European Union: An empirical analysis based on real-time data. Journal of Comparative Economics, 36(2), 221-242.
Guillamón, M. D., Bastida, F., & Benito, B. (2013). The electoral budget cycle on municipal police expenditure. European Journal of Law and Economics, 36(3), 447-469.
Vergne, C. (2009). Democracy, elections and allocation of public expenditures in developing countries. European Journal of Political Economy,25(1), 63-77.
Mink, M., & De Haan, J. (2006). Are there political budget cycles in the euro area?. European Union Politics, 7(2), 191-211.
Cusack, T. R. (1997). Partisan politics and public finance: Changes in public spending in the industrialized democracies, 1955–1989. Public choice, 91(3-4), 375-395.
Comiskey, M. (1993). Electoral competition and the growth of public spending in 13 industrial democracies, 1950 to 1983. Comparative Political Studies, 26(3), 350-374.
- The authors must refer to the limitations presented by the work. For example, they do not take into account the political orientation of the governing party. Besides, the do not consider the obligations public deficit what to be achieved by EU countries. Consider:
Buti, M., Eijffinger, S. C., & Franco, D. (2003). Revisiting the Stability and Growth Pact: grand design or internal adjustment?. Instituto Latinoamericano y del Caribe de Planificación Económica y Social, Santiago de Chile.
Buiter, W. H. (2006). The ‘Sense and Nonsense of Maastricht’revisited: What have we learnt about stabilization in EMU?. JCMS: Journal of Common Market Studies, 44(4), 687-710.
- After the analysis done, what recommendations could the authors suggest for policy economic?
Finally, once the work has been check, it can be publish.
Author Response
This paper is looking into interesting and current topic, since the management of the crisis Covid-19 of 2020 is going to influence a lot on the public expenditure of the countries.
This works has good arguments and show a complete statistical analysis.
I believe that the article needs to be thoroughly revise to allow for this:
-Do not appear an analysis of the independent variable: “government expenditures”, allow knowing its evolution between 2001-2012 and checking if there are important differences in the behaviour of this variable in the different countries, which could be explain by the independent variables.
A: We have provided a chart with analysis of dependent variable in appendix F with a comment on page 7, when discussing the variable.
-The paper analyses some determinants of “government expenditure”, NO all. This must be review.
A: We have added the needed qualification on page 8, when discussing the relative superiority of our method to test similar theories simultaneously.
-The hypothesis must be considering in section 2 “literature review and research hypotheses”. Taking into account the theoretical arguments that support them. Can not appear in the section 3 “Materials and Methods”.
A: We agree with that suggestion. The hypotheses were moved to the section 2 of the paper.
-The hypothesis 3 is not enough complete in “2.1.Institutions of Power Legitimacy”. You must be complete.
A: We are not sure if we understand the reviewer correctly, but actually as many as four hypotheses connect to the problem reviewed in section 2.1, H1, H2, H3 and H7. Moreover, if the problem is that the hypothesis 3 is stated in the negative, we stand by this decision, due to majority of research on the PBC at the country level supporting the rejection of PBC as useful theory in the context of last decades. We decided to leave it as it is (besides moving it to the section 2.
-The section “3.3 BMA-Bayesian model averaging”, you have to reduce it. Because the objective of this work is not show a develop mathematics model.
A: From the methodological section formulas 5, 7, 8 and 9 were eliminated, as well as the text relating to them. Those formulas were not essential to understanding of the estimation procedure and the information covered by those equation was left for references. Elimination of any other formula would distort the logical structure of this section and was retained as a result.
-You have to explain better the selection of countries considered in the analysis. It is not clear. Why these countries, specifically? Furthermore, Bulgaria is Not a member of the OECD. Consult:
http://www.oecd.org/about/members-and-partners/
A: thank you for this comment. We have now expanded the explanation of the choice of countries on page 7 and tested the model without Bulgaria – the results have not changed. The information on the results of this additional testing has been added to the section on robustness checks on page 16. The results of the analysis excluding Bulgaria can be seen below
Variable |
PIP |
PM |
PSD |
P(+) |
ER_nat |
1,000 |
-3,236 |
0,426 |
0,000 |
BBR_nat |
1,000 |
3,988 |
0,477 |
1,000 |
English |
1,000 |
17,382 |
1,740 |
1,000 |
French |
1,000 |
12,994 |
1,479 |
1,000 |
Socialist |
1,000 |
13,648 |
1,076 |
1,000 |
aaa_imf_rev |
1,000 |
0,405 |
0,078 |
1,000 |
aaa_gol_est |
1,000 |
4,118 |
0,562 |
1,000 |
EU |
1,000 |
6,730 |
1,268 |
1,000 |
CU |
1,000 |
-8,117 |
0,637 |
0,000 |
bmr_demdur |
1,000 |
0,068 |
0,009 |
1,000 |
checks |
1,000 |
-1,512 |
0,157 |
0,000 |
cl |
1,000 |
-6,882 |
0,603 |
0,000 |
Advanced |
1,000 |
6,323 |
0,592 |
1,000 |
resour_rich |
1,000 |
17,136 |
1,753 |
1,000 |
aaa_al_religion |
1,000 |
-12,382 |
1,381 |
0,000 |
aaa_al_ethnic |
1,000 |
-25,103 |
3,072 |
0,000 |
German |
1,000 |
11,396 |
1,618 |
1,000 |
aaa_dpi_gf |
1,000 |
7,848 |
1,935 |
1,000 |
DR_nat |
0,999 |
-2,121 |
0,455 |
0,000 |
deficit_pub |
0,999 |
-0,733 |
0,156 |
0,000 |
aaa_al_language |
0,987 |
9,958 |
2,794 |
1,000 |
lpart |
0,865 |
-0,824 |
0,437 |
0,000 |
aaa_h_polcon5 |
0,854 |
-7,436 |
4,093 |
0,000 |
aaa_dpi_mdmh |
0,596 |
0,028 |
0,027 |
1,000 |
x2009 |
0,578 |
-0,679 |
0,701 |
0,000 |
DR |
0,398 |
-0,629 |
0,919 |
0,000 |
deficyt_bud |
0,366 |
-0,112 |
0,172 |
0,000 |
RR_nat |
0,350 |
-0,434 |
0,687 |
0,000 |
Inf_IMF |
0,170 |
0,021 |
0,057 |
1,000 |
aaa_imf_gdpgr |
0,149 |
-0,011 |
0,035 |
0,125 |
Federal |
0,129 |
-0,150 |
0,598 |
0,173 |
BBR |
0,128 |
0,116 |
0,452 |
0,865 |
aaa_imf_ue |
0,102 |
-0,006 |
0,026 |
0,011 |
voter_turnout |
0,090 |
-0,001 |
0,007 |
0,159 |
Pub_debt |
0,086 |
-0,001 |
0,005 |
0,039 |
x2010 |
0,085 |
-0,033 |
0,169 |
0,000 |
OPEN |
0,079 |
0,008 |
0,045 |
1,000 |
aaa_h_polcon3 |
0,064 |
0,037 |
0,423 |
0,858 |
yrcurnt |
0,061 |
-0,001 |
0,023 |
0,177 |
aaa_dpi_legelec |
0,055 |
-0,001 |
0,059 |
0,423 |
- The conclusions obtained do not correspond to those derived from other works, such as those detailed below, which you must be consider it.
Galli, E., & Rossi, S. P. (2002). Political budget cycles: the case of the Western German Länder. Public choice, 110 (3-4), 283-303.
Giuliodori, M., & Beetsma, R. (2008). On the relationship between fiscal plans in the European Union: An empirical analysis based on real-time data. Journal of Comparative Economics, 36(2), 221-242.
Guillamón, M. D., Bastida, F., & Benito, B. (2013). The electoral budget cycle on municipal police expenditure. European Journal of Law and Economics, 36(3), 447-469.
Vergne, C. (2009). Democracy, elections and allocation of public expenditures in developing countries. European Journal of Political Economy,25(1), 63-77.
Mink, M., & De Haan, J. (2006). Are there political budget cycles in the euro area?. European Union Politics, 7(2), 191-211.
Cusack, T. R. (1997). Partisan politics and public finance: Changes in public spending in the industrialized democracies, 1955–1989. Public choice, 91(3-4), 375-395.
Comiskey, M. (1993). Electoral competition and the growth of public spending in 13 industrial democracies, 1950 to 1983. Comparative Political Studies, 26(3), 350-374.
A: Thank you for the useful references. They allowed us to broaden the discussion on PBC provided on page 3 and offer some discussion on the choice of government fractionalization as a competition variable on page 5 in the footnote.
- The authors must refer to the limitations presented by the work. For example, they do not take into account the political orientation of the governing party.
A: A short discussion on the absence of political ideology has been added to the conclusions/discussion section on page 18.
Besides, the do not consider the obligations public deficit what to be achieved by EU countries. Consider:
Buti, M., Eijffinger, S. C., & Franco, D. (2003). Revisiting the Stability and Growth Pact: grand design or internal adjustment?. Instituto Latinoamericano y del Caribe de Planificación Económica y Social, Santiago de Chile.
Buiter, W. H. (2006). The ‘Sense and Nonsense of Maastricht’revisited: What have we learnt about stabilization in EMU?. JCMS: Journal of Common Market Studies, 44(4), 687-710.
A: Thank you for this two references. We have used them in the discussion on the effectiveness of stability and growth pact on p. 14.
- After the analysis done, what recommendations could the authors suggest for policy economic?
A: The recommendations were added in the conclusions section, pp. 17-18
Finally, once the work has been check, it can be publish.
Reviewer 2 Report
Large and appropriate set of references, belonging to both political economy and economics.
Econometric analysis technically well constructed, adopting a relatively new approach.
Results quite reasonable, although not to surprising and in line with the existing literature.
Apart from some minor suggestions reported in the attached file, I would suggest also to test the role of economic globalization (trade openness, for example) in encouraging public spending, along the lines of what suggested, among others, by Dani Rodrik in several contributions. This aspect should be at least mentioned, if not added to integrate the present version.

Author Response
Large and appropriate set of references, belonging to both political economy and economics.
Econometric analysis technically well constructed, adopting a relatively new approach.
Results quite reasonable, although not to surprising and in line with the existing literature.
Apart from some minor suggestions reported in the attached file, I would suggest also to test the role of economic globalization (trade openness, for example) in encouraging public spending, along the lines of what suggested, among others, by Dani Rodrik in several contributions. This aspect should be at least mentioned, if not added to integrate the present version.
we added a proxy for the impact of globalization – as suggested by the reviewer we used trade openness as a proxy: exports and imports as a share of GDP. The variable turned out fragile with PIP equal to 0.123. The results are shown in the table below.
Variable |
PIP |
PM |
PSD |
P(+) |
BBR_nat |
1,000 |
3,466 |
0,532 |
1,000 |
English |
1,000 |
23,839 |
1,499 |
1,000 |
French |
1,000 |
15,859 |
1,301 |
1,000 |
Socialist |
1,000 |
14,421 |
1,292 |
1,000 |
aaa_h_polcon5 |
1,000 |
-20,947 |
2,548 |
0,000 |
aaa_imf_rev |
1,000 |
0,771 |
0,070 |
1,000 |
aaa_gol_est |
1,000 |
5,890 |
0,478 |
1,000 |
Federal |
1,000 |
-5,784 |
1,139 |
0,000 |
CU |
1,000 |
-8,046 |
0,764 |
0,000 |
checks |
1,000 |
-1,749 |
0,171 |
0,000 |
cl |
1,000 |
-4,633 |
0,668 |
0,000 |
Advanced |
1,000 |
5,070 |
0,733 |
1,000 |
aaa_dpi_gf |
1,000 |
7,760 |
1,364 |
1,000 |
German |
1,000 |
17,597 |
1,895 |
1,000 |
resour_rich |
1,000 |
8,691 |
1,937 |
1,000 |
ER_nat |
1,000 |
-2,184 |
0,428 |
0,000 |
bmr_demdur |
0,998 |
0,037 |
0,008 |
1,000 |
aaa_dpi_mdmh |
0,987 |
-0,101 |
0,024 |
0,000 |
aaa_al_religion |
0,970 |
-6,455 |
1,855 |
0,000 |
aaa_al_ethnic |
0,911 |
-6,201 |
2,649 |
0,001 |
deficyt_bud |
0,886 |
-0,586 |
0,308 |
0,000 |
EU |
0,879 |
-2,512 |
1,274 |
0,000 |
deficit_pub |
0,643 |
-0,307 |
0,290 |
0,000 |
aaa_imf_ue |
0,368 |
-0,048 |
0,074 |
0,003 |
Pub_debt |
0,200 |
-0,005 |
0,013 |
0,000 |
DR |
0,180 |
-0,199 |
0,520 |
0,006 |
aaa_al_language |
0,145 |
-0,877 |
2,911 |
0,035 |
DR_nat |
0,124 |
-0,075 |
0,256 |
0,003 |
OPEN |
0,123 |
0,024 |
0,081 |
1,000 |
yrcurnt |
0,119 |
-0,018 |
0,062 |
0,000 |
lpart |
0,113 |
-0,050 |
0,188 |
0,000 |
voter_turnout |
0,105 |
0,003 |
0,014 |
0,996 |
aaa_h_polcon3 |
0,099 |
0,196 |
0,796 |
0,999 |
aaa_dpi_legelec |
0,093 |
0,028 |
0,127 |
1,000 |
Inf_IMF |
0,084 |
0,006 |
0,034 |
0,997 |
BBR |
0,065 |
0,004 |
0,218 |
0,372 |
RR_nat |
0,063 |
0,021 |
0,203 |
0,898 |
x2009 |
0,060 |
-0,014 |
0,138 |
0,034 |
aaa_imf_gdpgr |
0,056 |
-0,001 |
0,012 |
0,234 |
x2010 |
0,051 |
0,000 |
0,113 |
0,502 |
As to the comments on the text:
Comment 1:
I am not sure this is the case. Not in EMU, at least.
We amended the sentence by adding: regardless of the legal constraints put on many countries in terms of their fiscal flexibility or some cases in which the crisis is connected directly to the level of public debt - such as in Greece
All other suggestions and comments made on the text were accepted. Authors are grateful for the help.
Round 2
Reviewer 1 Report
The paper has improved after changes made
Author Response
Thank you for accepting the changes made to the paper.